# Type I toxin-antitoxin systems contribute to the maintenance of mobile genetic elements in *Clostridioides difficile*

Johann Peltier[1,2], Audrey Hamiot[1,7], Julian R. Garneau [3], Pierre Boudry[1,8], Anna Maikova [1,2,4,9], Eliane Hajnsdorf [5], Louis-Charles Fortier [3], Bruno Dupuy[1] & Olga Soutourina [1,2,6✉]

Toxin-antitoxin (TA) systems are widespread on mobile genetic elements and in bacterial chromosomes. In type I TA, synthesis of the toxin protein is prevented by the transcription of an antitoxin RNA. The first type I TA were recently identified in the human enteropathogen *Clostridioides difficile*. Here we report the characterization of five additional type I TA within phiCD630-1 (*CD0977.1*-RCd11, *CD0904.1*-RCd13 and *CD0956.3*-RCd14) and phiCD630-2 (*CD2889*-RCd12 and *CD2907.2*-RCd15) prophages of *C. difficile* strain 630. Toxin genes encode 34 to 47 amino acid peptides and their ectopic expression in *C. difficile* induces growth arrest that is neutralized by antitoxin RNA co-expression. We show that type I TA located within the phiCD630-1 prophage contribute to its stability and heritability. We have made use of a type I TA toxin gene to generate an efficient mutagenesis tool for this bacterium that allowed investigation of the role of these widespread TA in prophage maintenance.

---

[1]Laboratoire Pathogenèse des Bactéries Anaérobies, CNRS-2001, Institut Pasteur, Université de Paris, 75015 Paris, France. [2]Université Paris-Saclay, CEA, CNRS, Institute for Integrative Biology of the Cell (I2BC), 91198 Gif-sur-Yvette, France. [3]Faculty of Medicine and Health Sciences, Department of Microbiology and Infectious Diseases, Université de Sherbrooke, 3201 rue Jean Mignault, Sherbrooke, QC J1E 4K8, Canada. [4]Center of Life Sciences, Skolkovo Institute of Science and Technology, Moscow 143028, Russia. [5]Institut de Biologie Physico-Chimique, UMR8261, CNRS, Université de Paris, 13 rue Pierre et Marie Curie, 75005 Paris, France. [6]Institut Universitaire de France (IUF), Paris, France. [7]Present address: UMR UMET, INRA, CNRS, Univ. Lille 1, 59650 Villeneuve d'Ascq, France. [8]Present address: Université Paris-Saclay, CEA, CNRS, Institute for Integrative Biology of the Cell (I2BC), 91198 Gif-sur-Yvette Cedex, France. [9]Present address: Peter the Great St. Petersburg Polytechnic University, Saint Petersburg 195251, Russia. ✉email: olga.soutourina@i2bc.paris-saclay.fr

Clostridioides difficile is a medically important human enteropathogen that became a key public health concern over the last two decades in industrialized countries[1,2]. This strictly anaerobic spore-forming Gram-positive bacterium is a major cause of antibiotic-associated nosocomial diarrhea in adults[3]. The main virulence factors of C. difficile are two toxins, TcdA and TcdB, produced by all toxigenic strains[4], and some isolates produce a binary toxin Clostridium difficile transferase (CDT). Additional factors, such as adhesins, pili, and flagella, involved in the interactions with the host during colonization have also been identified[5]. However, many questions remain unanswered regarding the success of this pathogen and its adaptation within the phage-rich gut environment.

C. difficile genome sequencing revealed the mosaic nature of its chromosome, which is composed of more than 10% of mobile genetic elements including integrated bacteriophages (prophages)[6]. Recent studies revealed a high prevalence of prophages in C. difficile genomes, each genome harboring between one and up to five prophages, either integrated into the chromosome or maintained as stable extrachromosomal circular DNA elements[7]. For example, the largely used laboratory strain 630 carries two homologous prophages, phiCD630-1 and phiCD630-2, while the NAP1/B1/027 epidemic strain R20291 carries one prophage (phi-027). The importance of prophages in the evolution and virulence of many pathogenic bacteria has clearly been demonstrated[8]. In C. difficile, all phages identified so far are temperate and can adopt a lysogenic lifecycle, and some of them have been shown to contribute to virulence-associated phenotypes. This includes modulation of toxin production and complex crosstalk between bacterial host and phage regulatory circuits[7–9]. When integrated into C. difficile genomes, prophages are stably maintained and replicated along with the host chromosome. However, when they are excised, either spontaneously or following induction by antibiotics or the exposure to other stress conditions, prophages can sometimes be lost during cell division and segregation. The rate of spontaneous phage loss under natural conditions has been estimated for Escherichia coli phages to range between $10^{-5}$ for phage P1 to $<10^{-6}$ for phage lambda[10,11]. To our knowledge, no experiments have been conducted to evaluate prophage loss rates in C. difficile.

TA modules are widespread in bacteria and archaea. These loci comprise two genes encoding a stable toxin and an unstable antitoxin[12]. Overexpression of the toxin has either bactericidal or bacteriostatic effects on the host cell while the antitoxin is able to neutralize the toxin action or production. For all identified TA modules, the toxin is always a protein. The RNA or protein nature and the mode of action of the antitoxin led to the classification of TA modules into six types[12]. In type I systems, the antitoxin is a small antisense RNA targeting toxin mRNA for degradation and/or inhibition of translation, while in type III systems, the antitoxin RNA binds directly to the toxin protein for neutralization[13,14]. For other TA types, both the toxin and the antitoxin are proteins. In most studied type II TA systems, the proteinaceous antitoxin forms a complex with its cognate toxin leading to toxin inactivation[15]. Major functions suggested for TA modules include plasmid maintenance, abortive phage infection and persistence, however, their role in persister cell formation in the presence of antibiotics remains a subject of controversy[16–26]. TA loci are commonly found on mobile genetic elements, in particular plasmids in which they were initially discovered and extensively studied. However, the roles of chromosomally-encoded TA modules, including those within prophage genomes, remain largely unexplored.

We recently reported the identification of the first type I TA systems associated with CRISPR arrays in C. difficile genomes[27]. The colocalization and coregulation by the general stress response

Sigma B factor and biofilm-related factors of TA, and CRISPR components suggested a possible genomic link between these cell dormancy and adaptive immunity systems. Interestingly, two of these functional type I TA pairs are located within the homologous phiCD630-1 and phiCD630-2 prophages in C. difficile strain 630. In the present work, we characterize additional type I TA modules highly conserved within C. difficile prophages, and provide experimental evidence of their contribution to prophage maintenance and stability. Moreover, we demonstrate here that inducible toxicity caused by type I toxins can be used as a counter selection marker in allele exchange genome editing procedures by promoting the elimination of plasmid-bearing cells, largely improving their efficiency.

## Results

**Identification of additional type I TA pairs in C. difficile.** Multiple TA modules have been discovered in bacterial chromosomes including prophage regions[12]. In C. difficile, we have recently identified several type I TA pairs adjacent to CRISPR arrays, two of them being located inside the phiCD630-1 and phiCD630-2 prophages of the strain 630 (CD0956.2-RCd10 and CD2907.1-RCd9, respectively)[27]. To determine whether other type I TA modules might be present within phiCD630-1, we performed a bioinformatics analysis on the phiCD630-1 sequence. Due to the small size of the toxin-encoding genes, standard methods of open reading frame (ORF) detection and gene annotation can hinder the identification of all toxin homologs. Moreover, prophages are characterized by a very high gene density, which can impede such detection of small and often overlapping coding regions. We therefore used the tBlastn program using the previously identified type I toxin CD0956.2 as a query, as it does not depend on annotation and ORF detection. We identified gene CD0977.1 and two other putative genes, unannotated on the genome, that we named CD0904.1 and CD0956.3. These genes code for small proteins of 47, 35, and 34 amino acids, respectively (Fig. 1a). Prophages phiCD630-1 and phiCD630-2 share a large region of homology with almost identical sequences, which include a duplication of CD0977.1 and CD0956.3 (named CD2889 and CD2907.2 in phiCD630-2, respectively) (Fig. 1b). In contrast, CD0904.1 is unique to phiCD630-1 and no other toxin gene homolog could be identified within phiCD630-2. Transcript reads were detected in regions of these putative genes by RNA-seq[28] (Supplementary Fig. 1a–c). The presence of a consensus RBS sequence (AGGAGG) 7–8 nucleotides upstream of the respective ATG start codons suggest that the corresponding proteins are produced. In addition, all three putative proteins carried a hydrophobic N-terminal region and a positively charged tail, which are characteristic features of type I toxins (Fig. 1a)[29]. Analysis of our previous TSS mapping data[28] and sequence alignments (Supplementary Fig. 1) suggested the presence of potential antisense RNAs of these toxin-encoding genes with the presence of TSS associated with Sigma A-dependent and Sigma B-dependent promoter elements for both the toxin and antitoxin genes (Supplementary Fig. 1)[30]. Antitoxins of CD0977.1, CD0904.1, and CD0956.3, located on phiCD630-1, were hereafter named RCd11, RCd13, and RCd14, respectively, and those of CD2889 and CD2907.2, found in phiCD630-2, were named RCd12 and RCd15.

To determine whether these additional potential TA pairs are functional, pRPF185-derivatives with anhydrotetracycline (ATc)-inducible $P_{tet}$ promoter were constructed to overexpress CD0904.1, CD0956.3, and CD0977.1 toxin genes (pT) or toxin–antitoxin modules (pTA) in C. difficile 630Δerm. Antisense RNAs are expressed from their own promoter in pTA. Growth of 630Δerm carrying the different pT and pTA vectors on BHI plates

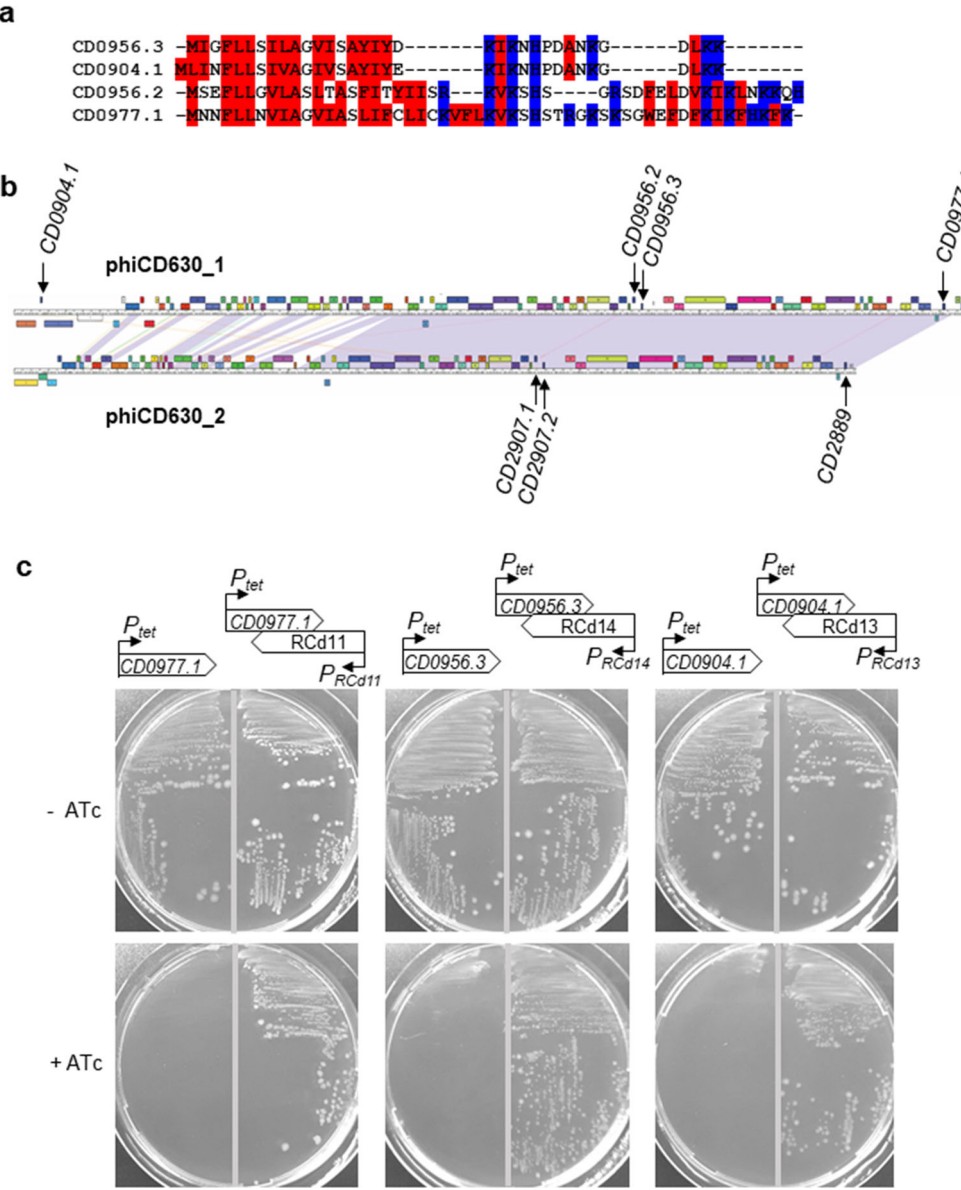

**Fig. 1 Identification and functionality of previously unannotated toxin genes within phiCD630-1. a** Protein alignment of toxin CD0977.1 with the newly identified CD0904.1, CD0956.2, and CD0956.3. The hydrophobic and positively charged amino acids are indicated in red and blue, respectively. **b** Maps and alignment of the phiCD630-1 and phiCD630-2 genomes. The location of toxin genes in both prophages is indicated. **c** Growth of *C. difficile* 630Δ*erm* strains harboring the pRPF185-based plasmids on BHI agar plates supplemented with Tm and with (+ATc) or without 10 ng/ml of ATc inducer (−ATc) after 24 h of incubation at 37 °C. Schematic representations of the constructs are shown.

was indistinguishable in the absence of ATc inducer (Fig. 1c). In contrast, growth of the 630Δ*erm*/pT strains was completely inhibited when ATc was present in the medium, while strains 630Δ*erm*/pTA showed a reversion of the growth defect. These results demonstrate that *CD0904.1*, *CD0956.3*, and *CD0977.1* encode potent toxins and are associated with antisense RNAs that function as antitoxins.

**Detailed characterization of the *CD0977.1*-RCd11 TA pair.** Intriguingly, predicted riboswitches responding to the c-di-GMP signaling molecule, cdi1_4 and cdi1_5, precede RCd11, and RCd12 antisense RNAs[28]. Most of these type I c-di-GMP-responsive riboswitches negatively control downstream genes by premature termination of transcription in the presence of c-di-GMP[28,31]. We therefore sought to further characterize the *CD0977.1*-RCd11 TA pair. In agreement with the data above,

addition of ATc to liquid cultures in exponential growth phase led to an immediate growth arrest of strain 630Δ*erm*/pT, unlike 630Δ*erm*/p (Supplementary Fig. 2a). In addition, the growth arrest was accompanied by a drop of colony-forming units (CFUs) (Supplementary Fig. 2b). Similarly to previous observations with other *C. difficile* type I TA modules[27], the analysis of liquid cultures by light microscopy showed that toxin over-expression was accompanied by an increase in cell length in about 10% of the cells (Supplementary Fig. 2c). Their length was above the mean length value of 630Δ*erm*/p control strain with two standard deviations (10.5 μm). The co-expression of the entire TA module led to the partial reversion of this phenotype.

Using Northern blotting, we detected both toxin and antitoxin transcripts in the 630Δ*erm*/p, 630Δ*erm*/pT (*CD0977.1*), and 630Δ*erm*/pTA (*CD0977.1*-RCd11) strains (Fig. 2a, b). In the absence of ATc inducer, a major transcript of about 300 nt was

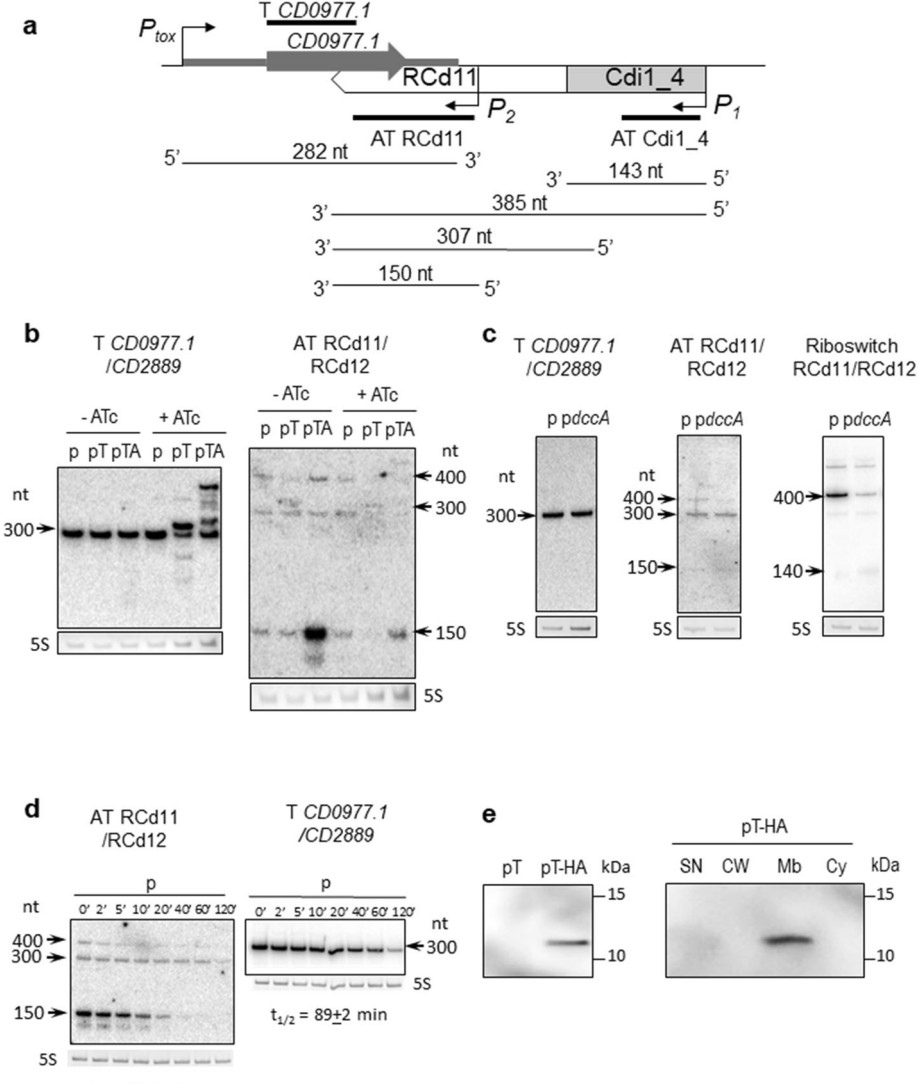

**Fig. 2 Detection of *CD0977.1* and RCd11 transcripts and CD0977.1-HA protein. a** A schematic of the *CD0977.1*-RCd11 TA pair genomic region and of the corresponding transcripts as identified by 5′/3′RACE and Northern blot. The Cdi1_4 riboswitch and the identified promoters are represented. The position of the different probes used in the Northern blot experiments is shown. **b** Northern blot of total RNA from *C. difficile* carrying p (empty vector), pT (expression of *CD0977.1*) or pTA (expression of *CD0977.1* and its antitoxin) in the absence (−ATc) or in the presence (+ATc) of 250 ng/ml of the inducer ATc. **c** Northern blot of total RNA from *C. difficile* carrying p (empty vector) or p*dccA* (expression of the diguanylate cyclase encoding gene *dccA*) in the presence of 250 ng/ml ATc. **d** Northern blot of total RNA from *C. difficile* 630Δ*erm* carrying an empty vector (wt/p) collected at the indicated time after addition of rifampicin. All Northern blots were probed with a radiolabelled oligonucleotide specific to the toxin (T CD0977.1/CD2889), the antitoxin (AT RCd11/RCd12) or the Cdi1_4/Cdi1_5 riboswitch (Riboswitch RCd11/RCd12) transcript and 5S RNA at the bottom serves as loading control. The arrows show the detected transcripts with their estimated size. The relative intensity of the bands was quantified using the ImageJ software. The half-lives for toxin and antitoxin transcripts were estimated from three independent experiments. **e** Detection and subcellular localization of the CD0977.1-HA protein. Immunoblotting with anti-HA detected a major polypeptide of ~12 kDa in whole cell extracts of *C. difficile* carrying pT-HA (CD0977.1-HA) grown in the presence of 250 ng/ml of ATc but not in extracts of *C. difficile* carrying pT (non-tagged CD0977.1) (left panel). The culture of *C. difficile* carrying pT-HA was fractionated into SN supernatant, CW cell wall, Mb membrane, and Cy cytosolic compartments and immunoblotted with anti-HA antibodies. Proteins were separated on 12% Bis–Tris polyacrylamide gels in MES buffer. See Supplementary Figs. 6 and 7 for uncropped Northern-blots and Western-blots, respectively.

detected in all three strains with a *CD0977.1*-specific probe. When using an RCd11-specific probe, transcripts of about 150, 300, and 400 nt were observed. Under inducing conditions, a reverse correlation between the relative toxin and antitoxin transcript abundance was noticed. The toxin overexpression in the presence of ATc inducer resulted in a decreased amount of the major 150-nt RCd11 antitoxin expressed from chromosomal location (lanes "pT" compared in the absence and in the presence of ATc). Similarly, for the strain carrying the entire TA locus on pTA

plasmid expressing the antitoxin from its own strong promoter, the toxin overexpression after ATc induction led to a decrease in the 150-nt RCd11 antitoxin level (lanes "pTA" compared under conditions "−ATc" and "+ATc"). To determine the impact of c-di-GMP on the antitoxin transcripts, we elevated c-di-GMP intracellular levels in the 630Δ*erm* wild type strain by expressing the gene *dccA*, coding for a diguanylate cyclase involved in c-di-GMP production, from a plasmid (p*dccA*) (Fig. 2c), as previously reported[28]. A c-di-GMP-regulated read-through transcript of

about 400 nt, as well as a terminated transcript of about 140 nt were detected in this strain by Northern blotting using a riboswitch-specific probe. In contrast, abundance of the 150-nt RCd11 antitoxin and toxin transcripts was not affected by fluctuations of c-di-GMP levels. Elevated c-di-GMP intracellular level could be associated with biofilm growth conditions. As for some other type I TA transcripts in our previous study[27], we detected by qRT-PCR analysis up to 20-fold increase in CD0977.1 toxin gene expression in biofilms as compared to planktonic culture ($20.4 + 5.0$, $N = 3$ biologically independent samples, with 16S rRNA gene for normalization), but no difference for short RCd11 form amount.

We then mapped the transcriptional start (TSS) and termination sites for the genes of the potential RCd11/RCd12-CD0977.1/CD2889 TA modules by 5′/3′RACE analysis (Supplementary Fig. 3 and Supplementary Table 1). The results obtained agreed well with the transcript lengths deduced from TSS mapping, RNA-seq, and Northern blot. Taken together, these data suggest the presence of two tandem TSS for RCd11, i.e., $P_1$ associated with c-di-GMP-dependent riboswitch, yielding a premature terminated transcript of ~140 nt, primary read-through transcript of ~400 nt (referred to as long transcript hereafter) and a processed transcript of ~300 nt, and $P_2$ located downstream from the riboswitch, yielding a transcript of ~150 nt (referred to as short transcript hereafter) (Supplementary Fig. 1). All these transcripts except for riboswitch-associated terminated transcript shared the same Rho-independent terminator (Supplementary Fig. 3). According to the position of the Northern blotting probes, the long 400-nt transcript could be detected with both riboswitch-specific and RCd11-specific probes, while terminated 140-nt transcript could be revealed only with riboswitch-specific probe and RCd11-specific probe hybridized to 300-nt and 150-nt transcripts (Fig. 2a).

We investigated the interaction between CD0977.1 toxin mRNA and the short and long RCd11 RNAs to determine, whether they form kissing complexes as in the case of the RCd9/CD2907.1 TA pair[27]. The results shown in Supplementary Fig. 4 reveal no difference in duplex formation with toxin mRNA for long and short antitoxin forms under native or full RNA duplex conditions suggesting that no kissing intermediate is formed during binding in native conditions in vitro. It should be noticed that only a fraction of CD0977.1 can interact with both antitoxin forms even when they are in excess, indicating that it is tightly folded in these experimental conditions (Supplementary Fig. 4).

It is in the nature of type I antitoxins to be short-lived in contrast to the stable toxin mRNA[14]. To determine the half-lives of toxin and antitoxin RNAs of the CD0977.1-RCd11 module, C. difficile strains were grown in TY medium until late-exponential phase and rifampicin was added to block transcription. Samples were taken at different time points after rifampicin addition for total RNA extraction and Northern blot analysis with toxin and antitoxin-specific probes. In a control strain 630Δerm/p carrying an empty vector, the half-life of the major short transcript for RCd11 was estimated to be about 8 min while the half-life of CD0977.1 toxin mRNA was estimated to about 89 min (Fig. 2d). Interestingly, depletion of the RNA chaperone protein Hfq, which generally increases the intracellular half-life of sRNAs and stabilizes the interactions between sRNAs and their target mRNAs, resulted in a moderate destabilization of CD0977.1 toxin mRNA and antitoxin RCd11 RNA with the half-life of 64 min and 4 min, respectively (Supplementary Fig. 5). By contrast, the stable CD0977.1 toxin mRNA was further stabilized to over 120 min half-life in the strains depleted for the ribonucleases RNase III, RNase J, and RNase Y, that could be involved in toxin and antitoxin RNA decay. For antitoxin RCd11 RNA, we also observed a stabilization in strain depleted for RNase Y

(Supplementary Fig. 5) suggesting that this ribonuclease contributes to antitoxin RNA degradation.

To confirm the protein nature of CD0977.1 and assess its subcellular localization, we constructed a derivative of CD0977.1 with an HA tag fused to the C-terminus of CD0977.1 expressed from a plasmid under the control of the inducible $P_{tet}$ promoter (pT-HA). C. difficile strain carrying pT-HA was grown to mid-exponential phase, induced with ATc for 90 min, and whole cell extracts were prepared. Induction of CD0977.1 expression immediately stopped the growth, as revealed by $OD_{600}$ measurements, suggesting that the HA-tag does not interfere with toxin activity. HA-tagged CD0977.1 was detectable by Western blotting with anti-HA antibodies (Fig. 2e). No band was observed in a whole cell extract of a control strain producing untagged CD0977.1 protein. The distribution of HA-tagged CD0977.1 within supernatant, cell wall, membrane, and cytosolic compartments was then studied (Fig. 2e). HA-tagged CD0977.1 was only detected in the membrane fraction, indicating the association of CD0977.1 with the cell membrane of C. difficile.

**The antitoxin transcript controlled by cdi1_4 riboswitch is dispensable for efficient toxin inactivation.** To get further insights into the function of abundant short (transcribed from $P_2$) and less abundant long RCd11 antitoxin transcripts (transcribed from $P_1$) (Supplementary Fig. 1), we generated plasmid constructs that allowed the inducible expression of the CD0977.1 toxin gene under the control of the $P_{tet}$ promoter and the expression of different forms of the RCd11 antisense RNA (Fig. 3a). The first construct, yielding pDIA6816, lacked the cdi1_4 riboswitch and its associated promoter ($P_1$) but retained the $P_2$ promoter of the antitoxin. On the opposite, the second construct, yielding pDIA6817, retained the $P_1$ promoter and the associated riboswitch but had a disrupted $P_2$ promoter. The construct in which both promoters of RCd11 and the riboswitch were present (pDIA6785), and the one in which only the toxin gene is expressed (pDIA6335) served as a positive and as a negative control for the assay, respectively. All plasmids were introduced into C. difficile 630Δerm and the corresponding strains were grown on BHI plates supplemented with 10 and 100 ng/ml ATc to induce CD0977.1 toxin expression. Growth of the strain carrying pDIA6816 was similar to that observed for the control strain carrying pDIA6785 in the presence of 10 ng/ml ATc and was slightly defective in the presence of 100 ng/ml ATc (Fig. 3a and Supplementary Fig. 8a). In contrast, the strain carrying pDIA6817 did not grow in the presence of 10 or 100 ng/ml ATc, similarly to the negative control strain. Similar results were obtained when the strains were grown in an automatic plate reader for 20 h in liquid medium in the presence of 5 ng/ml ATc (Fig. 3a). Interestingly, induction of toxin expression on BHI plate with a lower dose of ATc (5 ng/ml) led to a partial reversion of the growth defect of the strain carrying pDIA6817 unlike the negative control strain (Supplementary Fig. 8a). To further investigate the promoters functionality, a series of promoter fragments fused to the phoZ reporter gene was created in the wild type strain, and alkaline phosphatase (AP) activity was measured. After 4 h of growth in TY broth, the $P_2$ promoter fragment exhibited a reporter activity 1.4-fold lower than that of the full length promoter region, comprising $P_1$ and $P_2$, while the AP activity from the $P_1$ promoter associated with the Cdi1_4 riboswitch fragment was 5.7-fold lower (Fig. 3b). This suggests that the $P_1$ promoter activity is weaker than that of $P_2$, providing a rationale for the incompetence of the long antitoxin form expressed from $P_1$ for toxin inhibition. Of note, the promoter fragment comprising $P_1$ and the disrupted $P_2$ retained low AP activity (Supplementary Fig. 8b), suggesting that the nucleotide

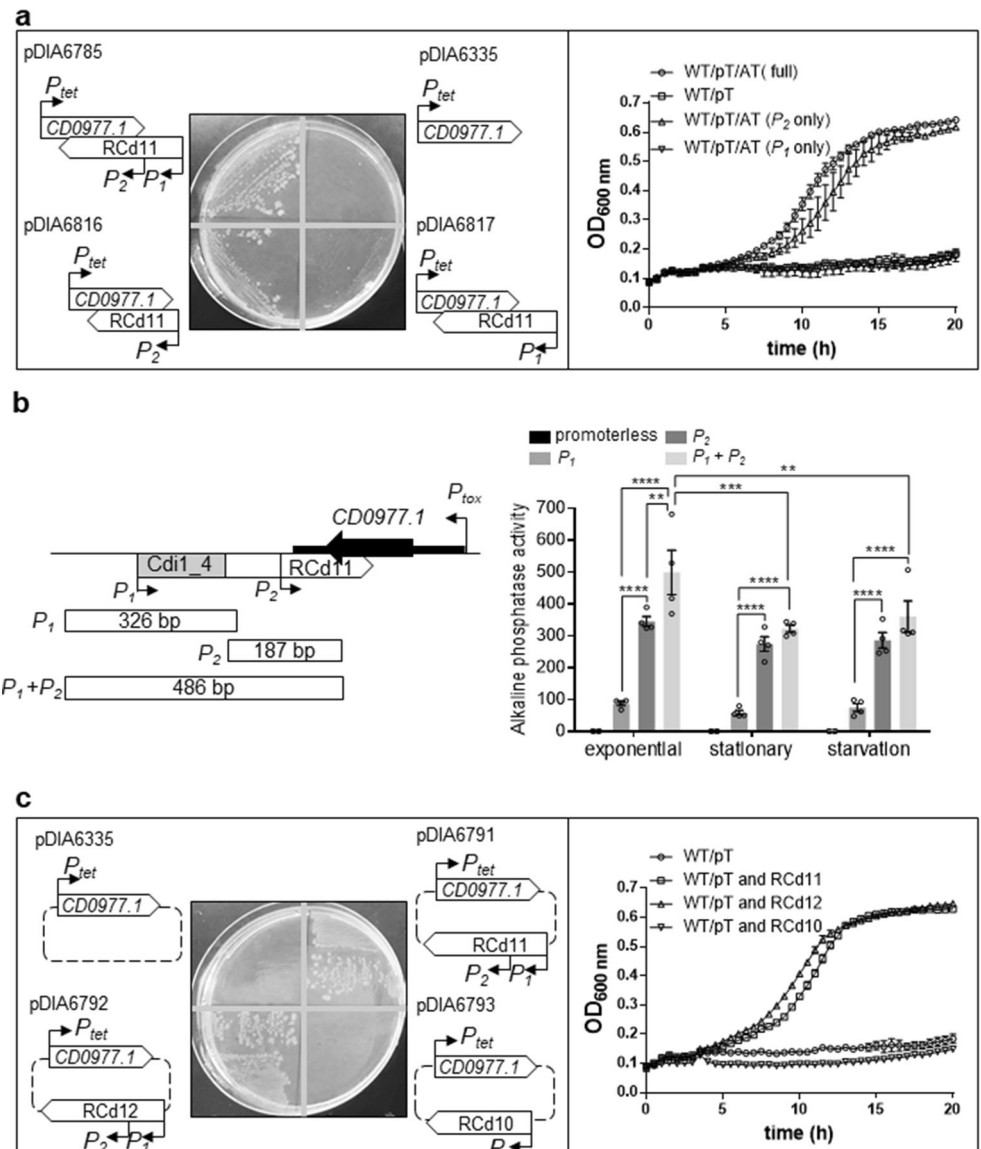

**Fig. 3 Impact of toxin-antitoxin co-expression on growth.** The effect on the toxicity of *CD0977.1* of long and short antitoxin transcripts expressed *in cis* (**a**) and *in trans* (**c**) was assessed. **a**, **c** Growth of *C. difficile* 630Δ*erm* strains harboring the pRPF185-based plasmids on BHI agar plates supplemented with Tm and 10 ng/mL of ATc inducer after 24 h of incubation at 37 °C and in TY broth at 37 °C in the presence of 5 ng/mL ATc. Schematic representations of the constructs are shown. Plotted values represent means and error bars represent standard error of the means ($N \geq 3$ biologically independent samples). **b** Alkaline phosphatase activity of the RCd11 promoter::*phoZ* reporter fusions measured after 4 (exponential) and 10 h (stationary) of growth in TY broth or under nutrient starvation conditions. A schematic of the *CD0977.1*-RCd11 TA pair genomic region and of the locations and sizes of promoter fragments constructed for the *phoZ* reporter fusions is shown. Values represent means and error bars represent standard error of the means ($N = 4$ biologically independent samples). **P ≤ 0.01, ***P ≤ 0.001, and ****P ≤ 0.0001 by a two-way ANOVA followed by a Dunnett's or Tukey's multiple comparison test. Source data are available in Supplementary Data 3.

substitutions in $P_2$ do not completely prevent expression of the antitoxin transcripts. Low level of antitoxin expression from $P_1$ promoter could thus explain the partial restoration of growth only in the presence of lower dose of ATc (5 ng/ml), when toxin is weakly expressed (Supplementary Fig. 8a). To determine whether promoters activity differs in various growth conditions, AP activity was next measured after 10 h of growth in TY broth and under nutrient starvation conditions. In these conditions, the full length promoter region ($P_1$ and $P_2$) exhibited a slight decrease in activity, while activity from the $P_1$ or $P_2$ promoter fusions was not significantly different to that observed during the exponential growth phase (Fig. 3b), suggesting that the expression of the antitoxin transcripts was not strongly modulated in these conditions. Taken together, these results suggest that the short

antitoxin transcript driven by promoter $P_2$ is crucial for the efficient inactivation of the toxin, while the longer antitoxin transcript directed by $P_1$ is dispensable.

**RCd12 counteracts toxic activity of noncognate CD0977.1 toxin.** Nucleotide sequences of short RCd11, lying within phiCD630-1 and short RCd12, lying within phiCD630-2, are almost identical with only three mismatches located near the 3′ end, in the region overlapping with the toxin transcript (Supplementary Figs. 9a and 10). The structure prediction suggested that the 3′ part folded similarly with two conserved hairpin structures in both antitoxin short and long form predictions (Supplementary Fig. 10). We therefore wondered whether RCd12

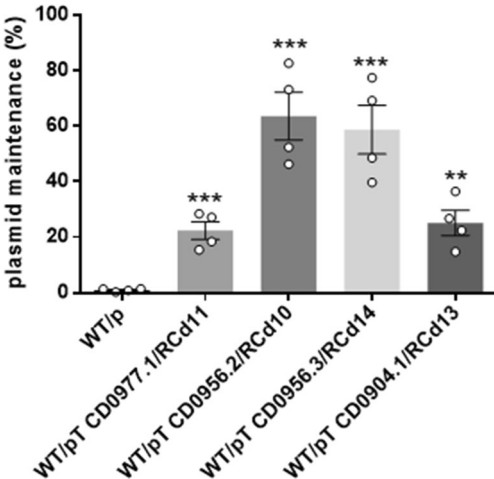

**Fig. 4 Impact of TA modules on plasmid loss in the absence of selection pressure.** The stability of pMTL84121 (p, empty vector) and pMTL84121-derived vectors expressing the different TA modules in *C. difficile* 630Δ*erm* was determined after seven passages (every 12 h) in TY broth without thiamphenicol. Values represent means and error bars represent standard error of the means ($N = 4$ biologically independent samples). **$P \leq 0.05$ and ***$P \leq 0.001$ by an unpaired *t*-test. Source data are available in Supplementary Data 3.

could crossreact with the transcript of the noncognate toxin CD0977.1. To answer this question, we generated constructs in which *CD0977.1* toxin gene under the control of the $P_{tet}$ promoter and different antitoxin genes with their own promoter were co-expressed from the same plasmid but from distant locations (Fig. 3c). As anticipated, expression of RCd11 in *trans* (pDIA6791) counteracted the toxicity associated with the expression of the cognate toxin both on plate and in liquid culture (Fig. 3c and Supplementary Fig. 8c). Replacement of RCd11 with RCd12 (pDIA6792) led to the same result (Fig. 3c and Supplementary Fig. 8c). By contrast, *in trans* expression of the more divergent RCd10 (the antitoxin of CD0956.2 toxin from a previously characterized TA module lying within phiCD630-1[27]) (pDIA6793) (Supplementary Fig. 9b) failed to revert the growth defect induced by *CD0977.1* expression (Fig. 3c and Supplementary Fig. 8c). These data indicate that antitoxins act in a highly specific manner to repress their cognate toxins, not only when they are expressed from the native convergent TA configuration, but also when expressed in *trans*. However, the specificity of interaction is permissive for at least three mismatches allowing RCd12 expressed from phiCD630-2 to efficiently prevent CD0977.1 toxin production from phiCD630-1.

**TA modules confer plasmid stabilization.** TA systems have been initially discovered on plasmids where they confer maintenance of the genetic element[18]. Plasmid loss results in a rapid decrease in the levels of the unstable antitoxin, which allows the stable toxin to inhibit cell growth. To test whether the TA modules located on phiCD630-1 could contribute to plasmid maintenance, we assessed the stability of pMTL84121-derived plasmids in which each TA module of phiCD630-1 was cloned and expressed under the control of their respective native promoter in *C. difficile* 630Δ*erm*. *C. difficile* 630Δ*erm* harboring the empty vector pMTL84121 was used as a control. After 7 passages in TY broth in the absence of antibiotic pressure, pMTL84121 was maintained by only 1.0% (+/−0.4%) of the bacterial population (Fig. 4). In contrast, plasmids expressing TA pairs were still present in 22.3 (+/−5.4%) to 63.6 % (+/−14.3%) of total cells. These results indicate that the four TA pairs can confer plasmid maintenance.

**Deletion of phiCD630-1 toxin genes in *C. difficile* 630Δ*erm*.** In order to determine whether the TA modules contribute to phiCD630-1 stability, we undertook the construction of mutants deleted for toxin genes of TA modules in *C. difficile* 630Δ*erm*. For this purpose, we constructed a modified Allele-Coupled Exchange (ACE) vector, derived from pMTL-SC7315, a *codA*-based "pseudosuicide" plasmid[32]. The *codA* cassette was here replaced with the *CD2517.1* toxin gene placed under the control of the $P_{tet}$ inducible promoter (Supplementary Fig. 11a). The functionality of RCd8-*CD2517.1* type I TA module in *C. difficile* was previously demonstrated[27]. In our vector, designated pMSR, the inducible toxic expression of *CD2517.1* is used as a counter-selection marker to screen for plasmid excision and loss (see "Methods" section), greatly facilitating the isolation of *C. difficile* deletion mutant generated by double cross-over allele exchange (Supplementary Fig. 11c, d). We also constructed a second vector, pMSR0, for allele exchange in *C. difficile* ribotype 027 strains and other ribotypes (see "Methods" section and Supplementary Fig. 11b). Using this improved tool, we first deleted the 49.3 kb phiCD630-2 locus to prevent any interfering cross-talk with phiCD630-1 (Supplementary Fig. 12). A multiple deletion mutant of toxin-encoding genes *CD0904.1, CD0956.2, CD0956.3,* and *CD0977.1* (phiCD630-1ΔT4) was then generated in the ΔphiCD630-2 background.

**TA systems are involved in maintenance of phiCD630-1 in the host cells.** Because the loss of an integrated phage from cells first requires its excision from the host genome, we sought to determine whether spontaneous excision of phiCD630-1 from chromosomal DNA occurred. To do so, we performed a PCR on genomic DNA from *C. difficile* 630Δ*erm* ΔphiCD630-2 with primers flanking the *attL* and *attR* sites of phiCD630-1 (Supplementary Fig. 13a). A PCR product with a size of 88 bp corresponding to a region with the excised prophage was detected (Supplementary Fig. 13b), and DNA sequencing of this amplicon confirmed the complete removal of phiCD630-1 from the host chromosome. A second PCR-based assay showed that the excised prophage (PCR product of 117 bp) was present as an extra-chromosomal circular form in the host cell (Supplementary Fig. 13a, b). We could also deduce the *attB, attP, attL,* and *attR* sites from the sequencing of the PCR products (Supplementary Fig. 11c). However, the frequency of phiCD630-1 excision, as measured by quantitative PCR (qPCR), was very low (~0.015%) (Supplementary Fig. 14). To screen for the presence/absence of phiCD630-1 in the host cells, we introduced, using our improved ACE vector, the *ermB* gene placed under the control of the strong *thl* promoter of *Clostridium acetobutylicum* as previously described[33], into an innocuous location (between *CD0946.1* and *CD0947*) of phiCD630-1 and phiCD630-1-ΔT4 (Fig. 5a). Starting from the overnight cultures, the ΔphiCD630-2 phiCD630-1::*erm* and ΔphiCD630-2 phiCD630-1-ΔT4::*erm* strains were subcultured four times in fresh medium, and cells were screened for erythromycin resistance by plating onto nonselective and erythromycin-containing agar plates. Nearly 100% of cells from both strains were found to still be resistant to erythromycin in these conditions, indicating that they had retained the prophage (Supplementary Fig. 13d).

In an attempt to artificially increase the excision rate of phiCD630-1, we ectopically expressed the putative excisionase *CD0912* of phiCD630-1 from the inducible $P_{tet}$ promoter, yielding pDIA6867. CD0912, identified in a bioinformatics search, is a 109 amino acid protein with a predicted DNA-binding domain similar to the HTH-17 superfamily and the excisionase (Xis) family[34]. Induction of *CD0912* expression with 10 ng/ml ATc in *C. difficile* 630Δ*erm* resulted in a high excision rate of phiCD630-

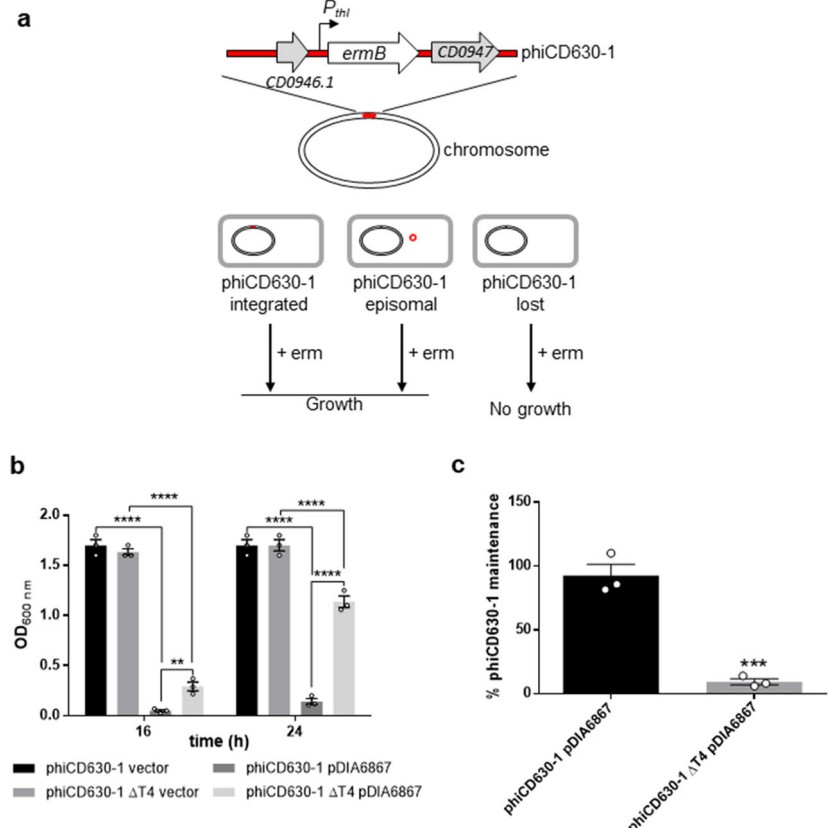

**Fig. 5 Impact of TA modules on prophage maintenance. a** Schematic representation of the method used to quantify prophage maintenance. A cassette containing an erythromycin resistance gene (*ermB*) under control of the strong *thl* promoter of *C. acetobutylicum* was introduced into an innocuous location of phiCD630-1, within the intergenic region between *CD0946.1* and *CD0947* genes, encoding a hypothetical protein and a putative scaffold protein, respectively. Cultures grown for 24 h were plated on erythromycin-containing agar plates and cells that lost the prophage were selectively killed. **b** Strains ΔphiCD630-2 phiCD630-1::*erm* and ΔphiCD630-2 phiCD630-1-ΔT4::*erm* carrying a vector control or pDIA6867 (overproducing the excisionase CD0912) were inoculated at an initial optical density at 600 nm ($OD_{600nm}$) of 0.005 in TY medium supplemented with 7.5 μg/ml Tm and 10 ng/ml ATc. Cultures were incubated at 37 °C and bacterial growth was determined by measurement of the $OD_{600nm}$ after 16 and 24 h. Values represent means and error bars represent standard error of the means ($N = 3$ biologically independent samples).\*\*$P \leq 0.01$ and \*\*\*\*$P \leq 0.0001$ by a two-way ANOVA followed by a Tukey's multiple comparison test. **c** Maintenance of prophages in strains ΔphiCD630-2 phiCD630-1::*erm* and ΔphiCD630-2 phiCD630-1-ΔT4::*erm* carrying pDIA6867 after 24 h of growth as in **b** was quantified by plating serial dilutions on agar plates supplemented or not with 2.5 μg/ml Erm. Values represent means and error bars represent standard error of the means ($N = 3$ biologically independent samples). \*\*\*$P \leq 0.001$ by an unpaired *t*-test. Source data are available in Supplementary Data 3.

1 (~70%), indicating that CD0912 functions as an excisionase for phiCD630-1 (Supplementary Fig. 14). Expression of *CD0912* in strains ΔphiCD630-2 phiCD630-1::*erm* and ΔphiCD630-2 phiCD630-1ΔT4::*erm* caused excision at a similar rate, suggesting that TA systems do not affect phiCD630-1 excision (Supplementary Fig. 14).

Strains ΔphiCD630-2 phiCD630-1::*erm* and ΔphiCD630-2 phiCD630-1-ΔT4::*erm* carrying pDIA6867 or a control vector were then grown in TY supplemented with 7.5 μg/ml Tm and 10 ng/ml ATc. After 16 and 24 h of incubation at 37 °C, measurement of the $OD_{600}$ revealed a dramatic growth defect of ΔphiCD630-2 phiCD630-1::*erm* expressing the excisionase compared to the strains carrying the control vector (Fig. 5b). Expression of the excisionase in ΔphiCD630-2 phiCD630-1-ΔT4:: *erm* also resulted in a growth defect, although at a lesser extent. In addition, plating of the cells bearing pDIA6867 on nonselective and erythromycin-containing agar plates revealed that phiCD630-1::*erm* was still present in more than 90% of the total population while phiCD630-1-ΔT4::*erm* remained in less than 10% of the cells (Fig. 5c). These results thus show that TA modules are important for phiCD630-1 maintenance after its excision and highlight the impact of the toxin expression on the

cell growth upon the loss of prophage. Together, these data demonstrate that TA modules contribute to phiCD630-1 heritability.

**Type I TA are prevalent in *C. difficile* phage genomes.** Since we identified additional toxin variants of type I TA systems after careful inspection of phiCD630-1 full genome, we decided to re-scan for possible ORFs in every available phage genomes of *C. difficile* using the permissive algorithm of the NCBI ORFfinder software. ORFs with minimal length of 60 nucleotides as well as nested ORFs were detected. A blastP search against the corresponding proteins allowed the identification of toxin homologs in all *C. difficile* prophage genomes (functional phages) (Fig. 6). Moreover, toxin sequence alignments revealed the high conservation of the hydrophobic *N*-terminal region, as well as the lysine-rich, positively charged region at the C-terminus. Hence, these data suggest the functionality of the toxins and reinforce their proposed role for phage maintenance and preservation.

Despite these conserved regions, alignment of toxins also revealed small variations among sequences. We therefore sought to explore the possible relationship between phage phylogeny and the observed toxin variants. A whole genome comparison of all

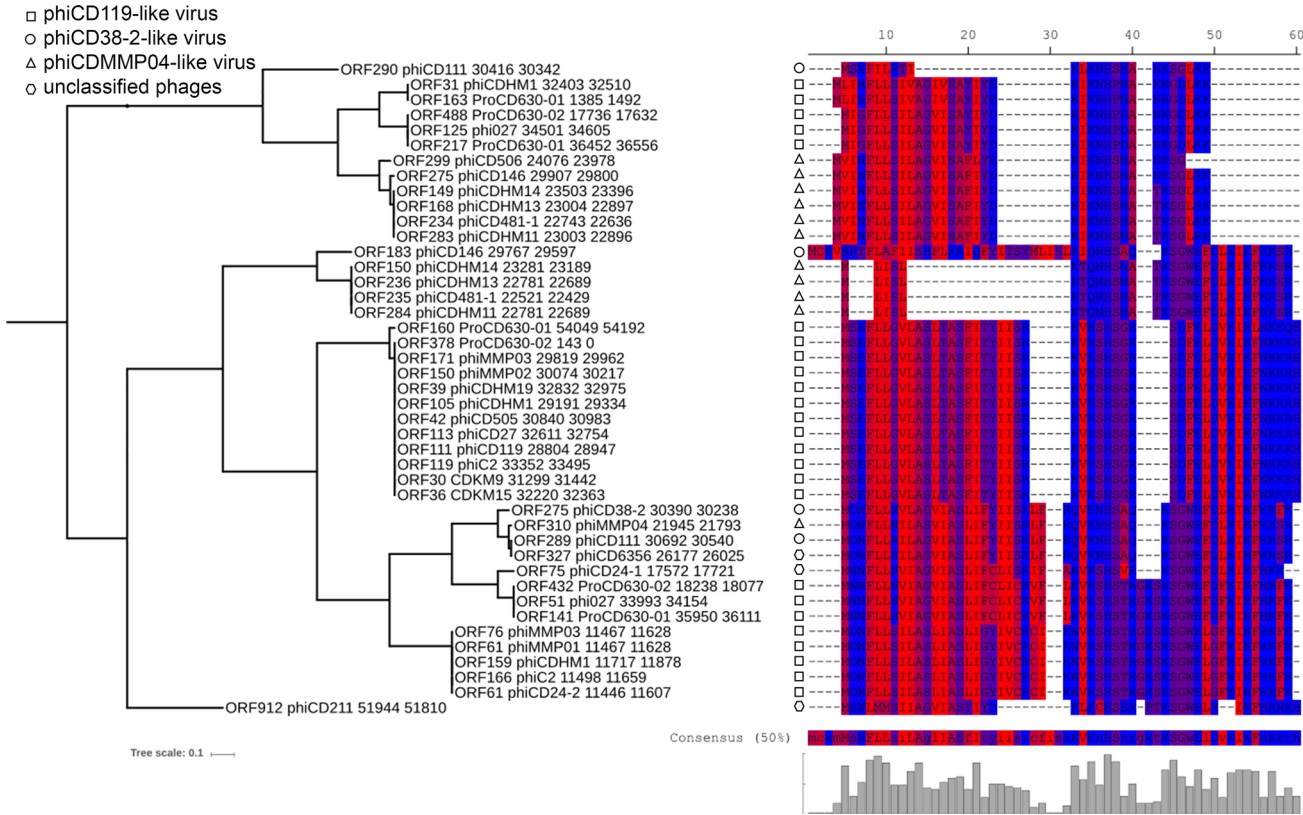

**Fig. 6 Relationship between phage phylogeny and toxin variants.** Putative toxin protein sequences detected in all available phage genomes were aligned using MUSCLE (v3.8) algorithm (EMBL-EBI). All phage genomes were re-scanned for potential ORFs using the NCBI ORFfinder software and detected ORFs were translated into their corresponding protein sequence. Protein sequences were combined to create a local BLASTp database and confirmed functional toxins CD0956.2 (large variant) and CD0904.1 (short variant) were used as queries to search for putative toxins in the database. Hits (min 45% identity, min 28% coverage) were retrieved and the corresponding proteins were aligned using MUSCLE (v3.8) algorithm (EMBL-EBI). The protein sequence consensus is shown. A phylogenetic tree was built using the Poisson distance method and neighbor joining implemented in Seaview (v4.4.2). Residues were colored according to high hydrophobicity (red) and low hydrophobicity (blue).

phages included in this study was performed to create phylogenetic groups (phiCD119-like viruses, phiCD38-2-like viruses, and phiMMP04-like viruses), as previously described[35]. A clear link between phage groups and specific toxin variants could be established, suggesting an independent acquisition of TA systems in different groups of phages (Fig. 6 and Supplementary Fig. 15). Interestingly, an extended search outside *C. difficile* phages revealed the presence of other toxin homologs inside plasmids of *C. difficile* and *Paeniclostridium sordellii*, a closely related species (Supplementary Fig. 16). These findings imply that *C. difficile* phages could recombine with plasmids to exchange genetic material, as already proposed for *E. coli* phages[36,37].

## Discussion

In this study, we identified and characterized several functional type I TA modules in *C. difficile* 630 prophages. Although these modules share characteristic features of known TA systems, i.e., (i) the toxins are membrane-associated proteins having a positively charged tail, (ii) the toxin mRNA is much more stable than the antitoxin RNA, (iii) artificial expression of the toxin genes inhibits bacterial growth unless their cognate antitoxin RNA is co-expressed; they do not present sequence homology with other TA modules identified to date in other bacteria.

RCd11-*CD0977.1* and RCd12-*CD2889* TA pairs are duplications respectively located within the homologous regions of phiCD630-1 and phiCD630-2 prophages. Two tandem TSS were identified for RCd11 (and RCd12), with the first one associated with the cdi1_4 (and cdi1_5) c-di-GMP-responsive riboswitch. C-

di-GMP is a second messenger in bacterial systems and a key signal in the control of critical lifestyle choices, such as the transition between planktonic and biofilm growth[28,38]. C-di-GMP has been found to regulate important functions in *C. difficile*, including motility, production of type IV pili, cell aggregation and biofilm formation, through control of gene expression by c-di-GMP-dependent riboswitches[38]. Sixteen predicted c-di-GMP sensing riboswitches are encoded in the *C. difficile* 630 genome and the regulatory function of five of them has been investigated so far. Cdi1_4 and cdi1_5 riboswitches were recently reported to be insensitive to an elevation of the c-di-GMP levels, and only transcript reads corresponding to the terminated transcript were detected, and no read-through seemed to occur[39]. However, our RACE-PCR and Northern-blot analysis indicated the presence of a transcript downstream from these riboswitches (Fig. 2, Supplementary Fig. 3 and Supplementary Table 1). Moreover, our data suggested that cdi1_4 and cdi1_5 are functional riboswitches responding to c-di-GMP since the abundance of the downstream transcripts was reduced in the presence of high levels of c-di-GMP (Fig. 2). In vitro interaction assays showed that both long and short RCd11 antitoxin RNAs could form a duplex with *CD0977.1* mRNA with the same efficiency, and only a fraction of toxin mRNA was included in these complexes probably due to extensive RNA folding (Supplementary Figs. 4 and 8). Despite these data, we found that the shorter and more abundant RCd11 transcript alone was sufficient to ensure complete CD0977.1 toxin inactivation under our conditions. In accordance, the analysis of promoter activities showed that c-di-

GMP-independent $P_2$ promoter driving the short RCd11 transcription was much stronger than the $P_1$ promoter. In contrast, the longer antitoxin transcript associated with the cdi1_4 riboswitch could only counteract the CD0977.1 toxicity when the toxin gene was expressed at low levels. This suggests that this antitoxin transcript might be involved in the tight regulation of CD0977.1 production and might be crucial to prevent toxin translation under conditions, where expression levels of the toxin gene would be slightly higher than those of the short antitoxin transcript. The c-di-GMP levels would then be critical in this regulation since elevated levels would result in a decreased abundance of the short RCd11 transcript and consequently in growth inhibition. From our previous studies of type I TA pairs, a larger link with biofilm-related control could be suggested for these TA systems since biofilm conditions affected the expression of several other TA transcripts independently from their association with c-di-GMP-responsive riboswitch[27].

In this work, we detected a natural background excision of the phiCD630-1 prophage and we identified the phage excisionase gene, CD0912. Expression of CD0912 from a plasmid promoted high levels of prophage excision from the host chromosome, mimicking prophage induction under stressful conditions. While phiCD630-1 and phiCD630-2 of C. difficile 630 share a large region of duplicated sequence, it is worth-noting that CD0912 is located in the variable region and has no homolog in phiCD630-2. Interestingly, no obvious putative excisionase-encoding gene could be identified in phiCD630-2 although natural excision of this prophage could also be detected in the course of our experiments. Moreover, expression of CD0912 had no impact on the excision rate of phiCD630-2, suggesting that phiCD630-2 might encode an atypical, yet to be identified excisionase. TA systems have been suggested to play three important biological functions, i.e., stabilization of mobile genetic elements (post-segregational killing), abortive phage infection and persister cell formation[40]. Prophage maintenance is among the suggested function of TA including a recent example of type II TA system that stabilizes prophage in Shewanella oneidensis[41], and another type II TA system promoting the maintenance of an integrative conjugative element in Vibrio cholerae[42]. This physiological function in prophage stabilization was also suggested for type I TA modules but had never been experimentally demonstrated prior to this study[43]. Prophage excision upon expression of CD0912 made phiCD630-1 prone to be lost by the host cells, and we could thus show that type I TA systems are important to maintain the episomal form of the phage into the host cell. In the C. difficile cells expressing the excisionase gene, the frequency of phiCD630-1ΔT4 loss was higher than that of wildtype phiCD630-1 and excision of phiCD630-1 was associated with a strong growth defect, which can be attributed to the post-segregational killing mechanism. The unstable antitoxin is likely degraded in daughter cells where the phage has been lost after cell division upon excision, leading to the toxin production from its stable mRNA and to the growth inhibition of the cell. Expression of the excisionase gene in cells carrying the prophage devoid of the toxin genes also resulted in a moderate growth defect, suggesting that a supplementary TA system might be present in phiCD630-1 or that the excisionase has an additional function affecting the cell growth. Several experimental conditions were tested in this work to induce the loss of the phage from the cells. Surprisingly, four passages of the strain carrying phiCD630-1 with the intact toxin genes grown in TY broth with constant expression of the excisionase gene from a plasmid resulted in approximately 99% of loss of this prophage. This is likely due to a progressive enrichment of the cell population surviving the loss of the phage, since the growth rate of this population is higher than that of the population bearing the phage. In any case, these data suggest that

the identification and the overexpression of the phage excisionase-encoding genes could provide an easy and efficient way to cure C. difficile strains from their prophages.

Toxins of type I TA systems are relatively small proteins, and this is probably one of the reasons why they have remained uncharacterized and unexplored in C. difficile and in other organisms. In this study, we have come to realize that standard methods of annotation are unable to detect all toxin homologs present in prophage genomes. Additional toxin homologs, previously unannotated, could thus be detected inside plasmids of C. difficile and P. sordellii. It has been proposed that phages could recombine with plasmids during infection of the same or different bacterial species to exchange genetic material[9,36,37]. It is therefore tempting to speculate that this TA system has the ability to disseminate, through horizontal gene transfer involving conjugation and recombination, from one species to another. Intriguingly, it was previously noticed that a 1.9-kb region could have been transferred from the plasmid of a C. difficile strain 630 to the phiCD38-2 prophage[9]. It was suggested that this recombination event had led to the acquisition of parA, a gene assumed to help the newly created chimeric phage to autonomously replicate and segregate as a circular plasmid. Our in silico search for TA systems in C. difficile phages reveals that this 1.9-kb region in phiCD38-2 also carried a TA (gp33) that presumably contributes to the phage maintenance and stability. It is interesting to observe that TA encoding regions can relocate from one mobile genetic element to another in this fashion, and that genes in proximity to the TA being transferred (i.e., parA gene) have more chances to become fixed in the newly integrated DNA. In the latter case, the region transferred seems to provide two complementary and beneficial features to the phage, i.e., the capacity to segregate successfully to the daughter cell, and the death of the cells upon curing if the phage has not been sequestered in both dividing cells. However, since TA systems behave as selfish elements that promote their propagation within bacterial genomes at the expense of their host[44,45], they are likely to be maintained and observed after their transfer by recombination events, even if they bring no selective advantage.

Thus, the large distribution of type I TA modules within C. difficile prophages argues in favor of their functional importance for prophage acquisition and transfer between C. difficile strains. The position of these TA modules in the extremities of the prophage is also consistent with their role in the entire prophage maintenance. In addition to prophage excision, other genetic events could lead to potential prophage loss such as recombination with other homologous phages. The presence of phage-like elements, cryptic phage rudiments and incomplete prophages in bacterial chromosomes including C. difficile genome attests on the frequency of such events. We could hypothesize that TA systems will contribute not only to episomal prophage stability, but also to the maintenance of integrated prophage.

Besides their biological functions, TA modules are also versatile tools for a multitude of purposes in basic research and biotechnology[46]. For example, the MazF toxin-encoding gene from E. coli is used as a counter-selection marker for chromosomal manipulation in Bacillus subtilis and C. acetobutylicum[47,48]. In this study, we engineered an inducible counter-selection marker based on the C. difficile CD2517.1 toxin gene of the CD2517.1-RCd8 TA module. Artificial expression of CD2517.1 from a plasmid in C. difficile leads to an immediate interruption of the bacterial growth[27]. Taking advantage of this feature, we generated improved vectors for allele exchange in C. difficile 630 (pMSR) and in C. difficile ribotype 027 strains and other ribotypes strains (pMSR0). It should be noted that expression of the RCd8 antitoxin from the pMSR0 vector was required to counteract the basal expression of CD2517.1 toxin gene due to the $P_{tet}$ leakiness. In

contrast, expression of the RCd8 antitoxin from the pMSR vector was not required since the *CD2517.1*-RCd8 TA module is naturally present within the chromosome of *C. difficile* 630. Native expression of RCd8 was therefore sufficient to prevent CD2517.1 production from the plasmid. Our vectors are derived from those developed by Cartman et al., which use the *codA* gene coding for cytosine deaminase as a counter-selection marker for allelic exchange mutations[32]. However, *codA*-based counter-selection was somewhat ineffective in our hands and false-positive counter-selected colonies with the plasmid still integrated into the chromosome were repeatedly found. This was reported by the authors as the consequence of loss-of-function mutations in genes leading to the bypass of the counter-selection. Our system proved to be much more efficient than all the others we have tested so far, and we did not observe any false-positive clones so far. The false-positive rate could be estimated to less than 0.1% since all counter-selected clones tested during about 50 mutant constructions attempts were thiamphenicol-sensitive, indicative of the plasmid loss. We successfully used this system to construct multiple mutants in various *C. difficile* strains, including the ΔT4 mutant, as well as deletion of a large chromosomal region of 50 kb corresponding to the phiCD630-2 prophage, and gene insertion into the bacterial chromosome (*ermB* gene). We therefore expect these improved vectors to become invaluable genetic tools that will foster research in *C. difficile*.

## Methods

**Plasmids, bacterial strains construction, and growth conditions**. *C. difficile* and *Escherichia coli* strains and plasmids used in this study are presented in Supplementary Data 1. *C. difficile* strains were grown anaerobically (5% $H_2$, 5% $CO_2$, and 90% $N_2$) in TY[49] or Brain Heart Infusion (BHI, Difco) media in an anaerobic chamber (Jacomex). When necessary, cefoxitin (Cfx; 25 µg/ml), cycloserine (Cs; 250 µg/ml), and thiamphenicol (Tm; 7.5 µg/ml) were added to *C. difficile* cultures. *E. coli* strains were grown in LB broth, and when needed, ampicillin (100 µg/ml) or chloramphenicol (15 µg/ml) was added to the culture medium. The nonantibiotic analog anhydrotetracycline (ATc) was used for induction of the $P_{tet}$ promoter of pRPF185 vector derivatives in *C. difficile*[50]. Strains carrying pRPF185 derivatives were generally grown in TY medium in the presence of 250 ng/ml ATc and 7.5 µg/ml Tm for 7 h, unless stated otherwise. Growth curves were obtained using a GloMax plate reader (Promega).

All primers used in this study are listed in Supplementary Data 2. Details of vector construction are described in the Supplementary Methods.

The resulting derivative plasmids were transformed into the *E. coli* HB101 (RP4) and subsequently mated with the appropriate *C. difficile* strains (Supplementary Data 1). *C. difficile* transconjugants were selected by subculturing on BHI agar containing Tm (15 µg/ml), Cfx (25 µg/ml), and Cs (250 µg/ml).

**Mutagenesis approach and mutant construction**. To improve the efficiency of the allele exchange mutagenesis in *C. difficile*, we made use of the inducible toxicity of the CD2517.1 type I toxin that we previously reported[27]. To construct the pMSR vector, used for allele exchange in *C. difficile* 630Δ*erm*, the *codA* gene was removed from the "pseudosuicide" vector pMTL-SC7315[32] by inverse PCR, and replaced by a 1169 bp fragment comprising the entire $P_{tet}$ promoter system and the downstream *CD2517.1* toxin gene. This fragment was amplified from pDIA6319 plasmid[27] and the purified PCR product was cloned into the linearized plasmid. In parallel, the pMSR0 vector, for allele exchange in *C. difficile* ribotype 027 strains and other ribotypes, was constructed by removing the *codA* gene from the vector pMTL-SC7215 by inverse PCR and replacing it with the *CD2517.1*-RCd8 TA region with *CD2517.1* under the control of the $P_{tet}$ promoter, as described above, and RCd8 under the control of its own promoter. For deletions, allele exchange cassettes were designed to have between 800 and 1050 bp of homology to the chromosomal sequence in both upstream and downstream locations of the sequence to be altered. The homology arms were amplified by PCR from *C. difficile* strain 630 genomic DNA (Supplementary Data 2) and purified PCR products were directly cloned into the PmeI site of pMSR vector using NEBuilder HiFi DNA Assembly. To insert $P_{thl}$-*ermB* into the phiCD630-1 prophage, within the intergenic region between *CD0946.1* and *CD0947* genes, homology arms (~900 bp upstream and downstream of the insertion site) were amplified by PCR from strain 630 genomic DNA (Supplementary Data 2). The $P_{thl}$-*ermB* cassette was amplified from the Clostron mutant *cwp19*[33,51]. Purified PCR products were all assembled and cloned together into the PmeI site of pMSR vector using NEBuilder HiFi DNA Assembly.

All pMSR-derived plasmids were initially transformed into *E. coli* strain NEB10β and all inserts were verified by sequencing. Plasmids were then transformed into *E. coli* HB101 (RP4) and transferred by conjugation into the appropriate *C. difficile* strains. The adopted protocol for allele exchange was similar to that used for the *codA*-mediated allele exchange[32], except that counter-selection was based on the inducible expression of the *CD2517.1* toxin gene. Transconjugants were selected on BHI supplemented with Cs, Cfx, and Tm, and then restreaked onto fresh BHI plates containing Tm. After 24 h, faster-growing single-crossover integrants formed visibly larger colonies. One such large colony was restreaked once or twice on BHI Tm plate to ensure purity of the single crossover integrant. Purified colonies were then restreaked onto BHI plates containing 100 ng/ml ATc inducer to select for cells in which the plasmid had been excised and lost. In the presence of ATc, cells in which the plasmid is still present produce CD2517.1 at toxic levels and do not form colonies. Growing colonies were then tested by PCR for the presence of the expected deletion.

**Light microscopy**. For light microscopy, bacterial cells were observed at 100× magnification on an Axioskop Zeiss Light Microscope. Cell length was estimated for more than 100 cells for each strain using ImageJ software[52].

**Subcellular localization of HA-tagged toxins by cell fractionation and Western blotting**. *C. difficile* cultures were inoculated from overnight grown cells in 10 ml of TY medium at an optical density at 600 nm ($OD_{600}$) of 0.05. Cultures were allowed to grow for 3 h before the addition of 250 ng/ml ATc and incubation for 90 min. Whole cell lysates were prepared using a single freeze-thaw cycle[50]. Briefly, cell pellets were harvested by centrifugation and frozen at −20 °C. Cells were thawed, resuspended in PBS containing 40 µg/ml DNase I to an $OD_{600}$ of 20, and incubated at 37 °C for 40 min. Culture supernatants were filtered with a 0.22 µm filter and precipitated on ice with 10% TCA for 30 min. Precipitates were harvested at max speed for 10 min at 4 °C, and the pellet was washed twice in ice-cold 90% acetone for 10 min. The pellet was finally resuspended in PBS to an $OD_{600}$ of 20. Cell fractionation was achieved using the CD27L endolysin[53]. Cultures of *C. difficile* were harvested by centrifugation and resuspended in phosphate/sucrose buffer (0.05 M $HNa_2PO_4$, pH 7.0, and 0.5 M sucrose) to an $OD_{600}$ of 20 with 30 µg/ml purified CD27L endolysin, and incubated at 37 °C for 1 h. Supernatants containing the cell wall fraction were retrieved, and the protoplast pellet was lysed with resuspension in phosphate buffer (0.05 M $HNa_2PO_4$, pH 7.0) containing 0.12 µg/ml DNase I at an $OD_{600}$ of 20 and incubated at 37 °C for 45 min. Lysates were harvested at max speed for 10 min at 4 °C, supernatants containing the cytoplasmic fraction were retrieved, and the membrane pellet was resuspended in phosphate buffer with 1% SDS at an $OD_{600}$ of 20. For analysis by SDS-PAGE, an equal volume of 2 × SDS sample buffer was added to protein samples. Coomassie staining was performed for loading and fractionation controls. Western blotting was performed with anti-HA antibodies (1:2000) (Osenses) using standard methods. Uncropped images of Western blots presented in the study are shown in Supplementary Fig. 7.

**Alkaline phosphatase activity assays**. *C. difficile* strains containing the *phoZ* reporter fusions were grown and harvested in exponential growth phase, at the onset of stationary phase and under nutrient starvation conditions. Starvation conditions correspond to a 1 h incubation of exponentially grown cells in PBS buffer at 37 °C in anaerobic conditions. Samples were stored at −20 °C and alkaline phosphatase assays were performed as detailed in the ref. [54] in the absence of chloroform. Cell pellets were washed with 0.5 ml of cold Wash buffer (10 mM Tris-HCl, pH 8.0, 10 mM $MgSO_4$), centrifuged and resuspended in 1 ml Assay buffer (1 M Tris-HCl, pH 8.0, 0.1 mM $ZnCl_2$). Five hundred microliter of the cell suspensions were transferred to separate tubes and mixed with 300 µl of Assay buffer and 50 µl of 0.1% SDS, before vortexing for 1 min. After incubation at 37 °C for 5 min, sample tubes were cooled on ice for 5 min. Samples were then incubated at 37 °C and 100 µl of 0.4% *p*NP (*p*-nitrophenyl phosphate in 1 M Tris-HCl, pH 8.0; Sigma-Aldrich) was added to each sample. A blank without cells was prepared as a negative control. The phosphatase reaction was stopped by addition of 100 µl of stop solution (1 M $KH_2PO_4$) and placing the tubes on ice. Time elapsed (min) for the assay was recorded. Assay samples were then centrifuged at max speed for 5 min and absorbance for each sample was read at both $OD_{420}$ and $OD_{550}$. Units of activity were calculated and normalized to cell volume by using the following formula: $((OD_{420} - (1.75 \times OD_{550})) \times 1000)/(t\ (min) \times OD_{600} \times vol.\ cells\ (ml))$.

**RNA extraction, Northern blot, and 5′/3′RACE**. Total RNA was isolated from *C. difficile* strains after 4, 6, or 10 h of growth in TY medium, or 7.5 h in TY medium containing 7.5 µg/ml of Tm and 250 ng/ml of ATc for strains carrying pRPF185 derivatives using Trizol (Sigma) (see ref. [55] for detailed protocol description). Starvation conditions corresponded to a 1 h incubation of exponentially grown cells (6 h of growth) in PBS buffer at 37 °C. Northern blot analysis and 5′/3′RACE experiments were performed as detailed in the ref. [28]. Five microgram of total RNA was separated on a denaturing 6 or 8% polyacrylamide gel containing 8 M urea, and transferred to Hybond-N + membrane (Amersham) by electroblotting. Following UV-cross-linking, prehybridization was carried out for 2 h at 42 °C in ULTRAHyb buffer (Ambion). Hybridization was performed overnight at 42 °C in the same buffer in the presence of a [γ-$^{32}$P]-labeled DNA oligonucleotide or PCR probe. After hybridization, membranes were washed twice for 5 min in 50 ml 2× SSC (300 mM sodium chloride and 30 mM sodium citrate) 0.1% sodium dodecyl

sulfate (SDS) buffer and twice for 15 min in 50 ml 0.1× SSC 0.1% SDS buffer. Radioactive signal was detected with a Typhoon system (Amersham). The size of the transcripts was estimated by comparison with RNA molecular weight standards (Invitrogen). The relative intensities of the bands from Northern blot analysis via autoradiography were quantified using ImageJ software. Uncropped images of Northern blots presented in the study are shown in Supplementary Fig. 6.

**RNA band shift assay**. Templates for the synthesis of RNA probes were obtained by PCR amplification using the Term and T7 oligonucleotides (Supplementary Data 2). The three RNAs (CD0977.1 toxin mRNA, the short and the long RCd11 antitoxin forms) were synthesized by T7 RNA polymerase and RNA concentrations were monitored by measuring the absorbance at 260 nm. Just before use, CD0977.1 was also transcribed with ($\alpha$-$^{32}$P) UTP yielding uniformly labeled RNA and traces were added to the unlabeled CD0977.1. This strategy was used because of the very low efficiency of 5′ labeling of all these transcripts. CD0977.1 transcript was incubated with increasing concentrations of RCd11 short or long RNAs under two different conditions referred as Native and Full RNA duplex conditions as in the ref. [27]. The complexes were immediately loaded on native polyacrylamide gels to control for hybridization efficiency. RNA levels were quantified by phosphoimagery.

**Measurement of RNA decay by rifampicin assay**. For determination of toxin and antitoxin RNA half-lives the C. difficile strains were grown in TY medium supplemented with 250 ng/ml ATc and 7.5 μg/ml Tm for 7.5 h at 37 °C. Samples were taken at different time points after the addition of 200 μg/ml rifampicin (0, 2, 5, 10, 20, 40, 60, and 120 min) and subjected to RNA extraction and Northern blotting.

**Plasmid stability assays**. Overnight cultures of C. difficile cells containing the pMTL84121 empty vector or the pMTL84121 derivatives were grown in TY broth with Tm and used to inoculate (at 1%) 5 ml of fresh TY broth without antibiotic. Every 10–14 h, 1% of the cultures were reinoculated into fresh TY broth without antibiotic. After seven passages, CFUs were estimated on TY plates supplemented or not with Tm to differentiate between the total number of cells and the plasmid-containing cells.

**Quantification of the frequency of prophage excision**. The frequency of prophage excision in different C. difficile strains was estimated by quantitative PCR on genomic DNA extracted using the NucleoSpin Microbial DNA kit (Macherey-Nagel). The total chromosome copy number was quantified based on the reference gene dnaF (CD1305) encoding DNA polymerase III. The number of chromosomes devoid of phiCD630-1 was quantified by PCR amplification using primers flanking phiCD630-1 (Supplementary Data 2), which only results in PCR products when the prophage is excised.

**PhiCD630-1 stability assays**. Overnight cultures of C. difficile strain 630 ΔphiCD630-2 phiCD630-1::erm and ΔphiCD630-2 phiCD630-1-ΔT4::erm were used to inoculate 10 ml of TY broth at an initial OD$_{600}$ of 0.05. Every 10–14 h, cultures were subcultured at an initial OD$_{600}$ of 0.05. After four passages, the cultures were serially diluted and plated on BHI plates to estimate the total CFUs and on BHI plates supplemented with 2.5 μg/ml erythromycin to determine the number of CFUs in which phiCD630-1 was still present. For cells expressing the excisionase gene, overnight cultures of C. difficile strain 630 ΔphiCD630-2 phiCD630-1::erm and ΔphiCD630-2 phiCD630-1-ΔT4::erm carrying pDIA6867 were used to inoculate fresh TY broth with Tm and 10 ng/ml ATc at an initial OD$_{600}$ of 0.005. After 24 h of incubation at 37 °C, cultures were serially diluted and plated on BHI plates to estimate the total CFUs and on BHI plates supplemented with 2.5 μg/ml erythromycin to determine the number of CFUs in which phiCD630-1 was still present. The OD$_{600}$ of the cultures was also measured to monitor cell growth.

**Statistics and reproducibility**. Data in graphs with error bars are presented as means ± standard error of the means (SEM) from at least three biologically independent experiments (n = 3). All Specific information on the number of replicates and statistical analyses are included in the figure legends. Data were processed with GraphPad Prism software.

**Reporting summary**. Further information on research design is available in the Nature Research Reporting Summary linked to this article.

## Data availability
All relevant data are available from the corresponding author upon reasonable request. Source data underlying plots shown in figures are provided in Supplementary Data 3. Full blots are shown in Supplementary Information. Newly generated plasmids pMSR and pMSR0 for chromosomal manipulation in Clostridioides difficile have been deposited in Addgene (78750).

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

## Acknowledgements

This work was supported by Agence Nationale de la Recherche ("CloSTARn", ANR-13-JSV3-0005-01 to O.S.), the Institut Universitaire de France (to O.S.), the University Paris-Saclay, the Institute for Integrative Biology of the Cell, the Pasteur Institute, the DIM-1HEALTH regional Ile-de-France program (LSP grant no. 173403), the CNRS-RFBR PRC 2019 (grant no. 288426, research project № 19-54-15003) to O.S., and a Vernadski fellowship to A.M., Centre National de la Recherche Scientifique (UMR8261), Université de Paris and the "Initiative d'Excellence" program from the French State (Grant "DYNAMO," ANR-11-LABX-0011 to E.H.). We would like to thank Shonna McBride for the gift of the pMC358 vector and Marc Monot for helpful discussions.

## Author contributions

J.P. and O.S. conceived and coordinated the study, which was initiated by P.B., J.P. and O.S. performed the majority of the experiments. A.H. constructed vectors and deletion mutants. J.R.G. performed the in silico analyses, A.M. performed growth curves and light microscopy. E.H. performed RNA band shift assays and provided insights on RNA interaction interpretations. L.-C.F. and B.D. provided scientific insight into the design of the experiments. J.P. and O.S. wrote the paper and all authors reviewed and approved the final version of the manuscript.

## Competing interests

The authors declare no competing interests.
