## [Peer Review File · Communications Biology]

Reviewers' comments:

Reviewer #1 (Remarks to the Author):

In this paper, Peltier *et al.* identify five prophage-encoded type I toxin-antitoxin (TA) systems in *Clostridioides difficile* and engineer a counter-selection mutagenesis method based upon a previously-characterised *C. difficile* TA system. They then employ their mutagenesis methodology to demonstrate the impact of the TA systems on prophage maintenance. The authors' methods are explained thoroughly and in sufficient detail to be able to reproduce the work.

The authors begin by identifying type I TA systems based upon BLAST homology to an existing TA system, CD0956.2, and then use classical TA-analysis methods to confirm that the components of the novel TA systems function as expected. The use of an inducible toxin alongside an antitoxin expressed from its native promoter on plasmids is well-supported methodology in the field, though more commonly for type II TA systems which autoregulate their expression based upon TA complex levels.

The authors go on to characterise CD0977.1-RCd11, confirming that expression of the toxin results in the expected cell lengthening that has been previously reported for type I TA systems in *C. difficile*. They also show levels of the transcripts via Northern blot, performing experiments that are consistent with there being a c-di-GMP-sensitive riboswitch on the RNA. The authors also characterise the half-lives of the RNA in the presence and absence of relevant chaperones and RNAses. Lastly, the authors use fractionation to localise CD0977.1 to the cell membrane. For the most part, this section of the paper is a comprehensive and clear characterisation of the TA system and I applaud the authors for their thoroughness. Whilst the mechanism of toxin activity is not elucidated, doing so would likely be outside of the scope of this manuscript given the work already contained within.

The authors' experiments to characterise the RCd10-12 transcripts and the associated antitoxicity with respect to toxin CD0977.1, as well as plasmid loss studies, are robust and well-presented. Using a combination of appropriate PCR and selection-based assays, the authors go on to find that the TA systems do not mediate excision of a prophage carrying them but do protect against the episome's loss from the cell with a dramatic post-segregational killing effect. There are some caveats and further findings within these experiments, which the authors address satisfactorily in the Discussion.

Lastly, the authors conduct a bioinformatic analysis of putative TA systems and their links to prophage families across *C. difficile* genomes. That they further extend their analysis outside of phages (to plasmids) is commendable and provides an interest avenue for future work.

The authors' development of a TA-dependent counter-selection mutagenesis protocol, whilst not an entirely novel concept, is likely to be important to the *C. difficile* field and may provide a significant advantage over existing mutagenesis methods.

Overall, this manuscript is well-written and its conclusions well-supported. As such, the following comments are ideas to be considered rather than fundamental flaws.

- Lines 129-131: The authors note that the toxins seem to contain similar regions and that this is "characteristic of type I toxins", but given that the hits were identified by BLAST homology to CD0956.2, this similarity between sequences seems unsurprising. Could the authors supply a reference or two for the "characteristic" statement?
- Lines 138-147: For type II TA systems, toxicity experiments would normally be carried out in strains that lack the associated antitoxin in its native locus, otherwise due to conditional

cooperativity there is an increase in antitoxin function that prevents toxicity. Presumably the copy number for type I TA systems is more relevant (e.g. a single-copy chromosomal antitoxin can't compete with a plasmid-encoded overexpressed toxin), but could the authors clarify whether this is indeed the case?

- Lines 166-167: This short section could be rewritten to better explain their point as it relates to the toxin and antitoxin levels.
- Line 258-269: For plasmid stability assays, was growth rate of the plasmid-bearing strains taken into account? The authors conduct this assay over "seven passages" every "10 to 14 hours" rather than a specific number of bacterial doublings. Whilst the results seem clear regardless, it would be worth stating whether or not the strains grew at the same rate in this assay and therefore whether any effect on generation time was negligible.
- Lines 309-318: This section presumably refers back to the PCR-based excision assay after introducing the reader to their erythromycin-based selection assay, which is a little confusing on first readthrough. For ease of understanding, perhaps the authors could reiterate in the legend for Figure S11 that it employs the PCR assay.
- Lines 465-468: How many clones have been tested in order to make the claim that no false positives were observed? It would be good for the authors to provide a vague idea of their false-positive rate based upon their detection limit (e.g. less than 1%, less than 0.01%, etc).

Reviewer #2 (Remarks to the Author):

Summary

This manuscript builds on previous work by the same authors in which they located (based on sequence homology) some type I TA systems in the vicinity of *C. difficile* CRISPR arrays and performed structure-function analysis on them (Maikova et al, ref. 21). This current work takes as its starting point those TA systems identified in close proximity to CRISPR arrays of the two prophages of *C. difficile* 630 and uses the known toxin sequence to interrogate the prophage genome sequences, resulting in the identification of five more systems of the same type. The authors characterize these systems and prove that they are functional toxin-antitoxin systems. They show that they can confer plasmid stabilization, using their standard overexpression plasmid as vector, and that they can maintain the prophage genome as an extrachromosomal genetic element under conditions of artificially induced excision from the genome, by overexpression of the prophage's excisionase gene. Finally, they show that one of the toxin genes can be used as a counterselection marker for construction of gene knockouts by allele exchange in *C. difficile*, analogous to the use of MazF as counterselection marker for this process in *E. coli*, and that in their hands this provides higher efficiency and lower false-positive rates compared to existing vectors used for this process in *C. difficile*. Bioinformatics data showing a high prevalence of TA systems of this type in other *C. difficile* prophages and plasmids are also provided.

The experiments are well performed, especially the Northern blots for toxin and antitoxin detection and stability analysis, and the data largely supports the authors conclusions. However, while the conclusion that the TA systems are capable of mobile genetic element maintenance per se is warranted, it is based on their ability to maintain an artificial cloning vector in the absence of antibiotic selection (Fig 4) and to maintain an artificially induced extrachromosomal form of the prophage (Fig 5C), which is not usually observed because these prophages normally remain integrated into the genome with (as the authors have measured themselves) a very low excision rate (Fig S11). It is therefore unclear how biologically relevant this role for these particular TA

systems will be, based on the data presented here, and since the role of other type I TA systems in mobile genetic element maintenance has been demonstrated in other bacteria many times, the novelty here is restricted to mobile genetic elements of *C. difficile* strains.

However, in my opinion the authors are on the edge of something highly novel with their discovery that the antitoxins of two of these systems are regulated by cyclic-di-GMP (cdG)-responsive riboswitches, and could very much increase the impact of their manuscript with a few experiments to pursue this further (see Major comments). Cyclic-di-GMP is a ubiquitous second messenger regulating many processes across the whole bacterial kingdom, and yet there is no existing literature in the field suggesting that it can regulate toxin-antitoxin system activity. If the authors can develop this further, it would certainly represent a major and highly influential step forward in the field.

Major comments

1. For the riboswitch-regulated antitoxins, as for some of the other type I antitoxins identified in the authors' previous work (ref 21) the antitoxins are present as short and long transcripts, where the longer one has a 5' extension due to its transcription from a more upstream promoter. However, there is a noticeable difference between the activity of the longer forms of the riboswitch-associated long transcripts in this manuscript and the authors' previous results for other long antitoxin transcripts. In their previous work (ref 21) both the short and long forms of antitoxin transcript could block toxin overexpression-induced growth inhibition, but in this manuscript the longer RCd11/12 antitoxin transcript from the upstream promoter, including the riboswitch, is incompetent for toxin inhibition (at 10 ng/ml ATc induction, Fig 3A). Since it overlaps 100% with the short form of the antitoxin and therefore contains all the complementary sequences needed for forming a complex with the toxin, this is quite unexpected and suggests that the long transcript containing the riboswitch is structured in a way that inhibits its interaction with the toxin mRNA. Since the long transcript is clearly downregulated by cdG (Fig 2B, consistent with the authors previous RNA-Seq work in ref 22) and can partially improve growth in the presence of low levels of overexpressed toxin (5 instead of 10 ng/ml ATc, Fig S5) it suggests that (1) the balance between toxin transcript, short antitoxin transcript and long antitoxin transcript is critical, and (2) TA system regulation by cdG could work by altering this balance. To investigate this, the authors could use the following experimental approaches:

- re-probe their +/- pdccA Northern blot of Fig 2B with the probe that detects the short transcript of the antitoxin
- perform in vitro RNA processing experiments as in their previous work (ref 21) to investigate whether the short or long transcripts show differences in RNase III processing with or without cdG and/or the toxin transcript
- make promoter-reporter constructs for the P1 and P2 promoters of the RCd11/RCd12 antitoxin(s) and investigate whether their activity differs in exponential or stationary phase or under conditions such as starvation vs. rich medium or biofilm vs. planktonic growth.
- indicate in Fig S7 whether the proposed structure here is for the longer or shorter form of the antitoxin transcript, and ideally include structure predictions for both long and short transcripts

It is not imperative that they perform all these experiments in all possible combinations, but data obtained from some or all of these experiments could significantly strengthen the manuscript.

2. It is well known that cdG levels can determine a switch between planktonic and biofilm associated lifestyles, with high cdG associated with biofilm and low cdG associated with motility or planktonic growth. In this context, it was very interesting to read in the authors' previous work (ref 21 again) that they observed some upregulation of transcripts of the type I TA system CD2907.1/RCd9 from the prophage phi630-2 in biofilm-grown cells. If the authors are still in possession of the biofilm and planktonic cDNA samples that were used for this qRT-PCR experiment, using them for qRT-PCR measurement of the transcripts of the new cdG-riboswitch-associated TA systems could provide a link between cdG regulation of these TA systems and the

growth mode of the cells. Careful qRT-PCR primer design could allow separate quantification of long transcript abundance, short transcript abundance and cdG-induced riboswitch processing from the same samples in biofilm or planktonic conditions.

3. While the authors' main conclusion (and the title) is that the TA systems can contribute to mobile genetic element maintenance, it is unclear how biologically relevant this is to the genetic elements (ie. the prophages) where they are naturally found. Unlike other temperate phages of *C. difficile* which can exist as episomes, the prophages phi630-1 and phi630-2 are integrated into the chromosome and have a very low natural excision rate. Therefore, the TA systems are not required for maintenance of the integrated prophage, and this should be stated more clearly. Lines 328-329 are not an appropriate conclusion here, because heritability of the prophage is largely assured by its natural stability in the genome.

4. Some data described in the Discussion is not found in the Results (lines 410-11, 414-17). It is stated in lines 410-11 that "Expression of the excisionase gene in cells carrying the prophage devoid of the toxin genes also resulted in a growth defect, although to a lesser extent". However, this data is not shown. Fig 5B only shows the results from excisionase overexpression with the wild type prophage or the prophage lacking TA systems, and there is no control with the empty vector of the excisionase overexpression plasmid to compare the effect of excisionase induction to. Moreover the data is only given as end-point OD600 measurements which give no indication of what phase of growth the cells are in. To allow proper evaluation of this experiment, it should include empty vector controls for both the wild type prophage and the $\Delta T4$ prophage and show the whole growth curves for these strains as in Fig 3.

In lines 414-17, it is stated that "four passages of the strain carrying phiCD630-1 with the intact toxin genes in TY broth with constant expression of the excisionase gene from a plasmid resulted in approximately 99% of loss of this prophage". This is not shown either, because the prophage maintenance experiment of Fig 5C only shows the maintenance after one passage of 24 hours, and this is expressed only as *erm* resistance which does not differentiate between maintenance in the chromosome or maintenance as circular extrachromosomal DNA. If this experiment could show data for both *erm* resistance and PCR-determined excision rates during up to 4 passages, it could give a lot more clarity on the relative timecourses of prophage excision vs. prophage loss and whether this is influenced by TA system presence or absence. At the moment it is hard to tell whether the authors are assuming that excision is immediately followed by loss, or whether the prophage is maintained extrachromosomally for some time.

Minor comments

Line 56, CDT? Undefined abbreviation

Line 60, missing reference for the *C. difficile* 630 original genome paper (Sebahia et al, 2006, Nat Genet)

Lines 75-77, clarify that these are the known loss rates for *E. coli* phages and specify whether anything is currently known about *C. difficile* phage loss

Line 81, neutralize the toxin action or production

Line 89, these references (refs 15-20) are not recent enough and also do not take into account the controversy regarding whether TA systems are really significant for persister cell formation. The authors should include a couple more references from labs who do not hold the same views (T Wood, L van Melderen labs, among others)

Lines 102-6, these two sentences are a non-sequitur with the rest of the introduction. Suggest removing the first sentence of this paragraph and beginning the second sentence with "Moreover" instead of "Thus" making it the final sentence of the previous paragraph.

Line 105, replace "improve" with "improving"

Line 119, suggest "does not depend on" in place of "overcome" and remove the word "defects"

Lines 166-67, "under inducing conditions, a reverse correlation between the relative toxin and antitoxin transcript abundance was noticed". It isn't clear what this reverse correlation is or what it

is reversed relative to, because the toxin transcript abundance appears the same in all lanes. Suggest re-writing for clarity.

Lines 183-87, suggest reformulating this sentence to clarify which probe hybridizes to which transcript

Line 245, remove "s" from "antitoxins"

Lines 428-29, can the authors elaborate why the TA system should have been passed from *P. sordelii* to *C. difficile* instead of the other way around? Are they found in the same environment? Is *P. sordelii* considered phylogenetically closer to the common ancestor?

Lines 435-37, this statement seems to be missing a reference, is this also from ref 8?

Figure 2 legend (lines 643-60), indicate more clearly in either the figure or the legend that the antitoxin is detected with the antitoxin probe in 2A but the riboswitch probe in 2B. For the stability experiments of Fig 2C, state in either the legend here or the materials and methods how many replicates the half-lives were calculated from.

Figure S3 legend, line 735 – "Potential processing site is shown by vertical green arrow" but there isn't a vertical green arrow in the figure, and line 739, "in blue boxes" instead of "underlined in blue" for consistency

Figure 3, indicate variation in the growth experiments by adding standard deviation error bars to the growth curves

Figure S4 legend (as for Fig 2C) state how many replicates the transcript half-lives were calculated from

Figure S7 legend, specify whether this is the short or long transcript (and ideally include structure prediction for short and long versions, as in major comment 1)

Figure S10D and 11, use correct Δ symbol in horizontal axis labels

Figure S13, what do the red dots by two of the toxin homologs signify?

Reviewer #3 (Remarks to the Author):

There are few comments that the authors can attempt to address in their revised manuscript.

1. For growth Curves, CFU analysis and light microscopy why only 1 TA pair (CD0977.1-RCd11) was chosen (Refer to S2ABC)

2. According to me CFU analysis data should be moved to main figures.

3. Author think 10 % longer cells can be significant number, Discussion lacks explanation behind how toxin expression can result in increased cell length. Also, why this phenomenon is only seen in 10% cells? Can the authors explain this?

4. The authors can shift bioinformatics analysis part before the results for characterization of Type I TA systems.

5. Can the authors functionally characterize few toxins (3-4) identified from bioinformatic work (fig. 6) in growth inhibition and cell length analysis.

6. Figure 3 legends: Line 666: How many experiments sets were done is missing

7. Axis label in Fig: 5C should be replaced with % phiCD630-1 maintenance.

8. The methods sections can be shortened.

9. The authors have referred to various strains which were published earlier. A brief explanation of these strains-a line each justifying their use would be beneficial for general audience.

Reviewer #4 (Remarks to the Author):

The manuscript by Peltier et al. describes the identification of new type I TA systems belonging to prophage regions of *C. difficile*. The authors show that toxins from these modules can be used as efficient counter-selectable marker for chromosome manipulation in *C. difficile*, and they provide the first evidence that type I TA systems could be involved in prophage maintenance.

Minor points:

- Toxin names such as CD0977.1 make the result and figure parts difficult to follow. It would be easier if the authors could simplify their nomenclature
- Fig 2A and 2B are not clear. How the RNA loads were adjusted (specially when the toxin was overexpressed)? What are the longer transcripts detected in pT and pTA lanes of Fig2A for the toxins? The relative abundance of transcripts seems to be gel-related. Are the transcripts detected with an antitoxin probe around 150 Nt Fig2A and 140 Nt Fig2B the same, and why results are so different with antitoxin probes between Fig2A and Fig2B? The authors could add a schematic to summarize the different transcripts starts and ends.
- page 7, lanes 166-167: The authors should clarify the reverse correlation they observe.
- For results presented in Fig3A, the authors could provide northern blots to show that mutated antitoxin transcripts are still expressed.
- Overall there are a lot of technical details in the results part that could be taken out for clarification.
- I do not understand why pMSR0 vector can be used for allele exchange in *C. difficile* 027

REVIEWERS' COMMENTS:

Reviewer #1 (Remarks to the Author):

I thank the authors for addressing my concerns thoroughly. I'm satisfied that my queries have been answered and that the manuscript has been modified appropriately to resolve any issues that I raised in my original review.

Reviewer #2 (Remarks to the Author):

Summary

The authors have revised their previously submitted manuscript on novel type I TA systems of *Clostridioides difficile* and their contribution to prophage maintenance with extra experiments, clarifying elements added to the figures and re-writing of sections of the text. Overall they have done an excellent job. The extra experiments, in particular the in vitro RNA interaction EMSAs and the promoter activity assays, significantly strengthen the manuscript. The schematics of northern blot probe positions and expected transcript lengths that they detect (Fig 2A) and promoter regions used for the alkaline phosphatase reporter assays (Fig 3B) are very helpful for interpretation of the results. The findings are important for the field and will certainly advance the state of the art regarding type I TA system function and regulation in general, not only in this species. I commend the authors for their efforts and have only a few minor comments, below.

Minor comments

1. line 183, c-di-GMP (not ci-)
2. Lines 185-87, the stated 20x differential expression of this toxin in biofilm-grown cells relative to planktonic by qRT-PCR is still in good support of the authors' conclusions about the biological relevance of the TA system even if no c-di-GMP effect was seen, and as such it merits showing the data. Not necessarily as a new figure, but in either a small table or directly in the main text including information on 1) how many replicates were performed, 2) average and standard deviation of the values if $n > 1$ and 3) whether it was normalized to a control transcript and if so, what was it. No objection to leaving the negative result for the antitoxin as "data not shown" statement.
3. Line 269-71, it is stated that the promoter-reporter containing P1 + disrupted P2 still retained AP activity, suggesting that the nucleotide substitutions in P2 (of the TSS and the -10 box, as per the supplementary methods) do not prevent expression of the antitoxin. But in Fig S6B this activity is only about 12 AP units. This is extremely low compared to the values in the graph of Fig 2B which look like about 80 units for P1 alone, 350 for P2 and maybe 500 units for P1 + P2 (in exponential growth). Moreover, comparing Fig 3A with FigS6B the pTA construct pDIA6817 containing the P_{tet}-toxin and the P1(+ disrupted P2) antitoxin is incapable of allowing growth on BHI-ATc plates with 10 or 100 ATc, but it can partially restore growth on 5 ATc. So this provides an explanation for why the growth of the strain containing pDIA6817 is partially restored on 5 ng/ml ATc (cf lines 259-61) – this low level of antitoxin expression from the P1 + disrupted P2 can neutralize the toxin when it is weakly expressed, but overwhelmed when more ATc inducer is included in the plates. Suggest rewriting for better synthesis of the results.
4. Minor inconsistency between legends of figure 3 and figure S6. Figure 3 legend states that 4 hr growth is exponential phase (line 719) and figure S6 legend states that 4 hr growth is stationary (line 830).

Reviewer #3 (Remarks to the Author):

The authors have provided appropriate responses to my comments. The manuscript should be accepted for publication

Reviewer #4 (Remarks to the Author):

The authors significantly improved the manuscript and they answered to all my comments.

Reply to comments of Reviewer 1

We would like to thank the Reviewer for the constructive criticism and helpful suggestions for improving the manuscript.

Reviewer #1 (Remarks to the Author): *In this paper, Peltier et al. identify five prophage-encoded type I toxin-antitoxin (TA) systems in Clostridioides difficile and engineer a counter-selection mutagenesis method based upon a previously-characterised C. difficile TA system. They then employ their mutagenesis methodology to demonstrate the impact of the TA systems on prophage maintenance. The authors' methods are explained thoroughly and in sufficient detail to be able to reproduce the work.*

The authors begin by identifying type I TA systems based upon BLAST homology to an existing TA system, CD0956.2, and then use classical TA-analysis methods to confirm that the components of the novel TA systems function as expected. The use of an inducible toxin alongside an antitoxin expressed from its native promoter on plasmids is well-supported methodology in the field, though more commonly for type II TA systems which autoregulate their expression based upon TA complex levels.

The authors go on to characterise CD0977.1-RCd11, confirming that expression of the toxin results in the expected cell lengthening that has been previously reported for type I TA systems in C. difficile. They also show levels of the transcripts via Northern blot, performing experiments that are consistent with there being a c-di-GMP-sensitive riboswitch on the RNA. The authors also characterise the half-lives of the RNA in the presence and absence of relevant chaperones and RNAses. Lastly, the authors use fractionation to localise CD0977.1 to the cell membrane. For the most part, this section of the paper is a comprehensive and clear characterisation of the TA system and I applaud the authors for their thoroughness. Whilst the mechanism of toxin activity is not elucidated, doing so would likely be outside of the scope of this manuscript given the work already contained within.

The authors' experiments to characterise the RCd10-12 transcripts and the associated antitoxicity with respect to toxin CD0977.1, as well as plasmid loss studies, are robust and well-presented. Using a combination of appropriate PCR and selection-based assays, the authors go on to find that the TA systems do not mediate excision of a prophage carrying them but do protect against the episome's loss from the cell with a dramatic post-segregational killing effect. There are some caveats and further findings within these experiments, which the authors address satisfactorily in the Discussion.

Lastly, the authors conduct a bioinformatic analysis of putative TA systems and their links to prophage families across C. difficile genomes. That they further extend their analysis outside of phages (to plasmids) is commendable and provides an interest avenue for future work.

The authors' development of a TA-dependent counter-selection mutagenesis protocol, whilst not an entirely novel concept, is likely to be important to the C. difficile field and may provide a significant advantage over existing mutagenesis methods.

Overall, this manuscript is well-written and its conclusions well-supported. As such, the following comments are ideas to be considered rather than fundamental flaws.

- *Lines 129-131: The authors note that the toxins seem to contain similar regions and that this*

is “characteristic of type I toxins”, but given that the hits were identified by BLAST homology to CD0956.2, this similarity between sequences seems unsurprising. Could the authors supply a reference or two for the “characteristic” statement?

Response :

As suggested by the Reviewer we have included a reference to Fozo et al. 2010 paper (line 134) describing the discovery of type I toxin families with characteristic protein features.

• *Lines 138-147: For type II TA systems, toxicity experiments would normally be carried out in strains that lack the associated antitoxin in its native locus, otherwise due to conditional cooperativity there is an increase in antitoxin function that prevents toxicity. Presumably the copy number for type I TA systems is more relevant (e.g. a single-copy chromosomal antitoxin can't compete with a plasmid-encoded overexpressed toxin), but could the authors clarify whether this is indeed the case?*

Response :

In our work, we used a high copy number toxin-overexpressing plasmid where the toxin-encoding gene has been placed under the control of the inducible *Ptet* promoter. Thus a single-copy chromosomal antitoxin gene could not be able to prevent the toxin-induced effect on growth.

• *Lines 166-167: This short section could be rewritten to better explain their point as it relates to the toxin and antitoxin levels.*

Response :

As requested by the Reviewers 1, 2 and 4, this statement has been clarified and additional details on the relative toxin and antitoxin levels have been included in the revised version of the paper as follows:

Lines 170-176: « The toxin overexpression in the presence of ATc inducer resulted in a decreased amount of the major 150-nt RCd11 antitoxin expressed from chromosomal location (lanes “pT” compared in the absence and in the presence of ATc). Similarly, for the strain carrying the entire TA locus on pTA plasmid expressing the antitoxin from its own strong promoter, the toxin overexpression after ATc induction led to a decrease in the 150-nt RCd11 antitoxin level (lanes “pTA” compared under conditions “-ATc” and “+ATc”).»

• *Line 258-269: For plasmid stability assays, was growth rate of the plasmid-bearing strains taken into account? The authors conduct this assay over “seven passages” every “10 to 14 hours” rather than a specific number of bacterial doublings. Whilst the results seem clear regardless, it would be worth stating whether or not the strains grew at the same rate in this assay and therefore whether any effect on generation time was negligible.*

Response :

No growth difference was observed between the plasmid-bearing strains. We did not state it in the manuscript because submission guidelines of the journal request that authors avoid « data not shown » statements.

• *Lines 309-318: This section presumably refers back to the PCR-based excision assay after introducing the reader to their erythromycin-based selection assay, which is a little confusing on first readthrough. For ease of understanding, perhaps the authors could reiterate in the*

legend for Figure S11 that it employs the PCR assay.

Response :

As suggested, the legend of Figure S11 (now Figure S12 in the revised version) has been completed as follows to specify that PCR-based excision assay has been used. Lines 868-869: «The frequency of prophage excision was estimated by quantitative PCR as described in Materials and Methods section. »

• *Lines 465-468: How many clones have been tested in order to make the claim that no false positives were observed? It would be good for the authors to provide a vague idea of their false-positive rate based upon their detection limit (e.g. less than 1%, less than 0.01%, etc).*

Response :

The claim that no false positives were observed is based on the fact that all counter-selected clones tested during our different mutant constructions attempts were thiamphenicol-sensitive, indicative of the plasmid loss. More than 50 mutants have been successfully constructed in the lab using our new strategy so far and we tested about ten clones per mutant. This information together with estimated false-positive rate of less than 0.1% has been included into the revised version of the manuscript. Lines 532-534: «The false-positive rate could be estimated to less than 0.1% since all counter-selected clones tested during about 50 mutant constructions attempts were thiamphenicol-sensitive, indicative of the plasmid loss. »

Reply to comments of Reviewer 2

We would like to thank the Reviewer for the constructive criticism and helpful suggestions for improving the manuscript.

Reviewer #2 (Remarks to the Author): This manuscript builds on previous work by the same authors in which they located (based on sequence homology) some type I TA systems in the vicinity of C. difficile CRISPR arrays and performed structure-function analysis on them (Maikova et al, ref. 21). This current work takes as its starting point those TA systems identified in close proximity to CRISPR arrays of the two prophages of C. difficile 630 and uses the known toxin sequence to interrogate the prophage genome sequences, resulting in the identification of five more systems of the same type. The authors characterize these systems and prove that they are functional toxin-antitoxin systems. They show that they can confer plasmid stabilization, using their standard overexpression plasmid as vector, and that they can maintain the prophage genome as an extrachromosomal genetic element under conditions of artificially induced excision from the genome, by overexpression of the prophage's excisionase gene. Finally, they show that one of the toxin genes can be used as a counterselection marker for construction of gene knockouts by allele exchange in C. difficile, analogous to the use of MazF as counterselection marker for this process in E. coli, and that in their hands this provides higher efficiency and lower false-positive rates compared to existing vectors used for this process in C. difficile. Bioinformatics data showing a high prevalence of TA systems of this type in other C. difficile prophages and plasmids are also provided.

The experiments are well performed, especially the Northern blots for toxin and antitoxin detection and stability analysis, and the data largely supports the authors conclusions. However, while the conclusion that the TA systems are capable of mobile genetic element maintenance per se is warranted, it is based on their ability to maintain an artificial cloning vector in the absence of antibiotic selection (Fig 4) and to maintain an artificially induced

extrachromosomal form of the prophage (Fig 5C), which is not usually observed because these prophages normally remain integrated into the genome with (as the authors have measured themselves) a very low excision rate (Fig S11). It is therefore unclear how biologically relevant this role for these particular TA systems will be, based on the data presented here, and since the role of other type I TA systems in mobile genetic element maintenance has been demonstrated in other bacteria many times, the novelty here is restricted to mobile genetic elements of C. difficile strains.

However, in my opinion the authors are on the edge of something highly novel with their discovery that the antitoxins of two of these systems are regulated by cyclic-di-GMP (cdG)-responsive riboswitches, and could very much increase the impact of their manuscript with a few experiments to pursue this further (see Major comments). Cyclic-di-GMP is a ubiquitous second messenger regulating many processes across the whole bacterial kingdom, and yet there is no existing literature in the field suggesting that it can regulate toxin-antitoxin system activity. If the authors can develop this further, it would certainly represent a major and highly influential step forward in the field.

Response to general criticism:

We agree with the Reviewer's criticism about the artificial nature of the experiments performed here to demonstrate the role of TA systems in prophage stabilization. However, even if the role of TA modules in the maintenance of mobile genetic elements has been largely suggested and shown for type II TA modules (we cited in the manuscript recent works on this role for type II TA), to our knowledge it was not yet experimentally demonstrated for the chromosomal type I TA.

Based on our experience, it is extremely difficult to cure *C. difficile* strains from prophages carrying active type I TA modules, suggesting their role in prophage maintenance. We agree that the rate of spontaneous prophage excision is rather low, however, the induction of prophages under stress conditions could lead to their loss in the absence of TA. We are mimicking here these conditions by artificial overexpression of the excisionase gene.

From an evolutionary point of view, the large distribution of type I TA modules within *C. difficile* prophages would also be in favour of their functional importance for prophage acquisition and transfer between *C. difficile* strains. Taken into account the importance of prophages for virulence in bacterial pathogens including Clostridia, this function should deserve a particular attention. However, this long-term effect on prophage stability and maintenance is difficult to demonstrate experimentally. The position of the TA systems in the extremities of the prophage is also in favour of their functional importance for the entire prophage maintenance. In addition to prophage excision, other genetic events could lead to potential prophage loss such as recombination with other homologous phages. The presence of phage-like elements, cryptic phage rudiments and incomplete prophages in bacterial chromosomes including *C. difficile* genome attests on the frequency of such events.

We also agree with the Reviewer about the novelty and the impact of a potential link between TA systems and c-di-GMP-dependent regulation. It is for this reason that these TA modules were selected for detailed analysis in our study. As suggested by the Reviewer, we have provided additional data to explore this aspect in more detail in the revised version of the manuscript to strengthen the work (see below).

Major comments

1. For the riboswitch-regulated antitoxins, as for some of the other type I antitoxins identified in the authors' previous work (ref 21) the antitoxins are present as short and long transcripts,

where the longer one has a 5' extension due to its transcription from a more upstream promoter. However, there is a noticeable difference between the activity of the longer forms of the riboswitch-associated long transcripts in this manuscript and the authors' previous results for other long antitoxin transcripts. In their previous work (ref 21) both the short and long forms of antitoxin transcript could block toxin overexpression-induced growth inhibition, but in this manuscript the longer RCd11/12 antitoxin transcript from the upstream promoter, including the riboswitch, is incompetent for toxin inhibition (at 10 ng/ml ATc induction, Fig 3A). Since it overlaps 100% with the short form of the antitoxin and therefore contains all the complementary sequences needed for forming a complex with the toxin, this is quite unexpected and suggests that the long transcript containing the riboswitch is structured in a way that inhibits its interaction with the toxin mRNA. Since the long transcript is clearly downregulated by cdG (Fig 2B, consistent with the authors previous RNA-Seq work in ref 22) and can partially improve growth in the presence of low levels of overexpressed toxin (5 instead of 10 ng/ml ATc, Fig S5) it suggests that (1) the balance between toxin transcript, short antitoxin transcript and long antitoxin transcript is critical, and (2) TA system regulation by cdG could work by altering this balance. To investigate this, the authors could use the following experimental approaches:

- re-probe their +/- *pdccA* Northern blot of Fig 2B with the probe that detects the short transcript of the antitoxin
- perform *in vitro* RNA processing experiments as in their previous work (ref 21) to investigate whether the short or long transcripts show differences in RNase III processing with or without cdG and/or the toxin transcript
- make promoter-reporter constructs for the P1 and P2 promoters of the RCd11/RCd12 antitoxin(s) and investigate whether their activity differs in exponential or stationary phase or under conditions such as starvation vs. rich medium or biofilm vs. planktonic growth.
- indicate in Fig S7 whether the proposed structure here is for the longer or shorter form of the antitoxin transcript, and ideally include structure predictions for both long and short transcripts

It is not imperative that they perform all these experiments in all possible combinations, but data obtained from some or all of these experiments could significantly strengthen the manuscript.

Response:

- re-probe their +/- *pdccA* Northern blot of Fig 2B with the probe that detects the short transcript of the antitoxin

We have re-probed the Northern blot with the probe that detects both short and long forms of RCd11 antitoxins and only the detection of the 385-nt long form was affected by c-di-GMP level modulation. Results are shown in Fig. 2C.

- perform *in vitro* RNA processing experiments as in their previous work (ref 21) to investigate whether the short or long transcripts show differences in RNase III processing with or without cdG and/or the toxin transcript

We have performed the *in vitro* experiments to investigate the interaction between CD0977.1 toxin mRNA and the short and long forms of antitoxin RCd11 RNAs. We observed no difference in duplex formation with toxin mRNA for long and short AT forms under native or full RNA duplex conditions suggesting that no kissing intermediate is formed during

binding in native conditions *in vitro*. It should be noticed that only a fraction of *CD0977.1* mRNA can interact with both antitoxin forms even when they are in excess, indicating that it is tightly folded in these experimental conditions. Results of this *in vitro* interaction analysis are shown in new Figure S4 and have been included in the results (lines 202 to 209) and discussion sections (lines 421 to 424).

We can assume that for this type of riboswitch the presence of c-di-GMP will mainly modify the RNA conformation to induce premature transcription arrest and generate a prematurely terminated 140-nt transcript that does not contain complementarity with toxin mRNA. In the absence of c-di-GMP the transcription continues and results in a long transcript that overlaps the toxin mRNA (385-nt long form).

The second short form of 150-nt antitoxin is transcribed from the P2 promoter independently of c-di-GMP and contains no site of interaction with c-di-GMP. Thus, *in vitro* interaction studies in the presence and in the absence of c-di-GMP are not relevant for the short form with respect to its mode of action. The only long form that can interact with c-di-GMP is present as a minor form in the presence of c-di-GMP.

- *make promoter-reporter constructs for the P1 and P2 promoters of the RCd11/RCd12 antitoxin(s) and investigate whether their activity differs in exponential or stationary phase or under conditions such as starvation vs. rich medium or biofilm vs. planktonic growth.*

As suggested by the Reviewer, we constructed alkaline phosphatase fusions to measure the P1 and P2 promoter activities under various conditions. As now shown in Fig. 3B, our results suggest that the activity of the P1 promoter is much weaker than that of P2 providing a rationale for the incompetence of the long antitoxin form for toxin inhibition. No significant difference was observed for the activity of the P1 and P2 promoters in the different conditions tested, including exponential and stationary phase as well as starvation. Results are presented in lines 261 to 276 of the result section and also mentioned in the discussion (lines 426 to 428). We also attempted to investigate the activity of the promoters in biofilm conditions but no activity could be detected suggesting that the cells were dead or metabolically inactive in our conditions (72 h incubation in rich medium). These latter data are not included in the manuscript.

- *indicate in Fig S7 whether the proposed structure here is for the longer or shorter form of the antitoxin transcript, and ideally include structure predictions for both long and short transcripts*

As suggested, we have included the structure prediction for both the short and long forms of the antitoxin in Figure S7 (new Figure S8 in revised version). The 3' part folded similarly with two conserved hairpin structures in both short and long form predictions. This information has been added in lines 283 to 285 of the revised manuscript: "The structure prediction suggested that the 3' part folded similarly with two conserved hairpin structures in both antitoxin short and long form predictions (Fig. S8)"

2. *It is well known that cdG levels can determine a switch between planktonic and biofilm associated lifestyles, with high cdG associated with biofilm and low cdG associated with motility or planktonic growth. In this context, it was very interesting to read in the authors' previous work (ref 21 again) that they observed some upregulation of transcripts of the type I TA system CD2907.1/RCd9 from the prophage phi630-2 in biofilm-grown cells. If the authors are still in possession of the biofilm and planktonic cDNA samples that were used for this*

qRT-PCR experiment, using them for qRT-PCR measurement of the transcripts of the new cdG-riboswitch-associated TA systems could provide a link between cdG regulation of these TA systems and the growth mode of the cells. Careful qRT-PCR primer design could allow separate quantification of long transcript abundance, short transcript abundance and cdG-induced riboswitch processing from the same samples in biofilm or planktonic conditions.

Response :

Despite careful qRT-PCR primer design, the complex structure of the analysed region prevented us from making clear quantification of all transcripts under the biofilm conditions in comparison with planktonic culture. We could observe up to 20-fold increase in *CD0977.1* toxin gene expression in biofilms and no difference for short AT form amount.

Among identified TA modules, only two TA analysed in this work are associated with c-di-GMP riboswitches. However, we have observed upregulation of other TA transcripts in biofilms in our previous work that could be related to additional mechanisms independent from the presence of c-di-GMP-responsive riboswitches that deserve further investigations. These results and considerations have been included into results (lines 183 to 187: «Elevated ci-di-GMP intracellular level could be associated with biofilm growth conditions. As for some other type I TA transcripts in our previous study, we detected by qRT-PCR analysis up to 20-fold increase in *CD0977.1* toxin gene expression in biofilms as compared to planktonic culture, but no difference for short RCd11 form amount (data not shown). » and discussion sections (lines 436 to 439) of the revised manuscript.

Altogether these new results strengthen our work demonstrating that

- the short RCd11 antitoxin form is transcribed from the stronger P2 promoter as compared to the P1 promoter driving the long c-di-GMP-riboswitch-associated form;
- both antitoxin RCd11 RNAs are similarly structured and able to form a duplex with *CD0977.1* toxin mRNA but only a fraction of toxin mRNA participates in these complexes under *in vitro* conditions

Thus, the main difference between short and long antitoxin forms resides within the level of their expression due to stronger P2 promoter activity under all conditions tested that could explain the differences in their toxin inactivation capacities.

*3. While the authors' main conclusion (and the title) is that the TA systems can contribute to mobile genetic element maintenance, it is unclear how biologically relevant this is to the genetic elements (ie. the prophages) where they are naturally found. Unlike other temperate phages of *C. difficile* which can exist as episomes, the prophages *phi630-1* and *phi630-2* are integrated into the chromosome and have a very low natural excision rate. Therefore, the TA systems are not required for maintenance of the integrated prophage, and this should be stated more clearly. Lines 328-329 are not an appropriate conclusion here, because heritability of the prophage is largely assured by its natural stability in the genome.*

Response :

First, if TA systems contribute to episomal prophage stability, it is reasonable to suggest that they will also reinforce the stability of integrated prophages. Even if loss rates of integrated prophages are lower than those of episomal prophages, integrated prophages can also certainly benefit from the protective effect of TA systems, for example by avoiding elimination by bacteria during the excised stage of their replication cycle. Second, TA systems might also act as protective elements to preserve the integrity of integrated prophages

inside the bacterial genome (ex: by protecting the prophage from deletions, genome rearrangements, and gradual decay (doi: 10.1128/mmbr.67.2.238-276.2003.)) In this regard, we have observed that TA systems seem to be well dispersed along some integrated prophage sequences (ex: TA at the upstream border, at the center and at the downstream border). This could possibly prevent damaging modifications by bacteria all along the prophage sequence. However, this would be long and difficult to show experimentally, since prophage decay happens over long time ranges. Furthermore, TA systems have been suggested to help protect and stabilize other large integrated genomic elements such as superintegron arrays (DOI: 10.1111/j.1365-2958.2007.05613.x), or other smaller integrated genetic elements such as cryptic prophages and conjugative transposons (DOI: 10.1371/journal.pgen.1000439). Finally, stabilization during prophage transfer between strains and dispersion of these elements within a population could be also suggested as an important point for TA function. The conclusion: “Together, these data demonstrate that TA modules mediate phiCD630-1 heritability » has been modulated and now reads : « Together, these data demonstrate that TA modules contribute to phiCD630-1 heritability. » (line 373). We have included these considerations into the Discussion section of the revised manuscript (lines 501 to 510).

4. Some data described in the Discussion is not found in the Results (lines 410-11, 414-17). It is stated in lines 410-11 that “Expression of the excisionase gene in cells carrying the prophage devoid of the toxin genes also resulted in a growth defect, although to a lesser extent”. However, this data is not shown. Fig 5B only shows the results from excisionase overexpression with the wild type prophage or the prophage lacking TA systems, and there is no control with the empty vector of the excisionase overexpression plasmid to compare the effect of excisionase induction to. Moreover the data is only given as end-point OD600 measurements which give no indication of what phase of growth the cells are in. To allow proper evaluation of this experiment, it should include empty vector controls for both the wild type prophage and the $\Delta T4$ prophage and show the whole growth curves for these strains as in Fig 3.

Response :

The data discussed here have now been added in the Results section. Controls with the empty vectors have been added to compare the effect of excisionase induction (lines 362 to 368). We have now added the time point 16 h to show that the strains are still growing between 16 and 24 h (see Fig. 5B).

In lines 414-17, it is stated that “four passages of the strain carrying phiCD630-1 with the intact toxin genes in TY broth with constant expression of the excisionase gene from a plasmid resulted in approximately 99% of loss of this prophage”. This is not shown either, because the prophage maintenance experiment of Fig 5C only shows the maintenance after one passage of 24 hours, and this is expressed only as erm resistance which does not differentiate between maintenance in the chromosome or maintenance as circular extrachromosomal DNA. If this experiment could show data for both erm resistance and PCR-determined excision rates during up to 4 passages, it could give a lot more clarity on the relative timecourses of prophage excision vs. prophage loss and whether this is influenced by TA system presence or absence. At the moment it is hard to tell whether the authors are assuming that excision is immediately followed by loss, or whether the prophage is maintained extrachromosomally for some time.

Response :

We have added “(data not shown)” at the end of this statement (line 471). While it would be interesting to investigate in detail the relative time-course of prophage excision vs prophage loss, we think that it lies outside the scope of this paper. Our point of view on this aspect, clarified in the Discussion (lines 462 to 465), is that the prophage is maintained extrachromosomally for some time after excision and can be lost during cell division. In presence of TA systems, this loss is followed by the production of toxin from the stable toxin mRNA, resulting in the poor growth observed in these conditions.

Minor comments

Response :

All the minor points raised by Reviewer were considered and the suggested modifications were made.

Line 56, CDT? Undefined abbreviation

Response:

Clostridium difficile transferase (CDT) is now defined (line 60).

Line 60, missing reference for the C. difficile 630 original genome paper (Sebaihia et al, 2006, Nat Genet)

Response:

The missing reference has been included (line 66).

Lines 75-77, clarify that these are the known loss rates for E. coli phages and specify whether anything is currently known about C. difficile phage loss

Response :

We have now mentioned more clearly that those loss rates are values for *E. coli*, but that to our knowledge, no experiments have been conducted to evaluate prophage loss rates in *C. difficile* (lines 80 to 82).

Line 81, neutralize the toxin action or production

Response :

The sentence has been modified according to Reviewer’s suggestion (line 86).

Line 89, these references (refs 15-20) are not recent enough and also do not take into account the controversy regarding whether TA systems are really significant for persister cell formation. The authors should include a couple more references from labs who do not hold the same views (T Wood, L van Melderens labs, among others)

Response :

As requested, we have now modulated the statement on the role of TA in persister cell formation and included the references from TK Wood and L van Melderens labs (lines 93 to 95).

Lines 102-6, these two sentences are a non-sequitur with the rest of the introduction. Suggest

removing the first sentence of this paragraph and beginning the second sentence with “Moreover” instead of “Thus” making it the final sentence of the previous paragraph.

Response :

As suggested the first sentence has been removed placing the second sentence as the final in the paragraph (line 107).

Line 105, replace “improve” with “improving”

Response :

Done (line 109)

Line 119, suggest “does not depend on” in place of “overcome” and remove the word “defects”

Response :

Done (line 122)

Lines 166-67, “under inducing conditions, a reverse correlation between the relative toxin and antitoxin transcript abundance was noticed”. It isn’t clear what this reverse correlation is or what it is reversed relative to, because the toxin transcript abundance appears the same in all lanes. Suggest re-writing for clarity.

Response :

As requested also by Reviewers 1 and 4, this statement has been clarified and additional details on the relative toxin and antitoxin levels have been included in the revised version of the paper as follows:

Lines 170-176: « The toxin overexpression in the presence of ATc inducer resulted in a decreased amount of the major 150-nt RCd11 antitoxin expressed from chromosomal location (lanes “pT” compared in the absence and in the presence of ATc). Similarly, for the strain carrying the entire TA locus on pTA plasmid expressing the antitoxin from its own strong promoter, the toxin overexpression after ATc induction led to a decrease in the 150-nt RCd11 antitoxin level (lanes “pTA” compared under conditions “-ATc” and “+ATc”).»

Lines 183-87, suggest reformulating this sentence to clarify which probe hybridizes to which transcript

Response :

To clarify the position of probes with respect to identified transcripts, we have now included the schematic representation of the different transcripts and the probes used for their detection in Figure 2A.

Line 245, remove “s” from “antitoxins”

Response :

Done

Lines 428-29, can the authors elaborate why the TA system should have been passed from P. sordelii to C. difficile instead of the other way around? Are they found in the same environment? Is P. sordelii considered phylogenetically closer to the common ancestor?

Response:

It has recently been suggested that *P. sordellii* is a sister group to *C. difficile*, as both clades were found to be monophyletic (doi: 10.1186/s12864-015-1663-5). This very close relationship makes it difficult to predict which species is the closest to the common ancestor, and also makes it difficult to speculate on the orientation of transfer of genetic elements between the two species. *C. difficile* and *P. sordellii* do not cause the same disease and pathology in humans, however, they both have shown the capacity to infect the digestive tract of common animals, such as felines, horses, cattle and sheep (DOI: 10.1354/vp.43-3-370s) (DOI: 10.1186/1471-2180-14-173). In these hosts, by sharing the same environment, they might have exchanged genetic material by conjugation or by infection by similar phages and exchange of genetic elements from *P. sordellii* to *C. difficile* has recently been suggested (DOI: 10.1128/mBio.01761-17). For the sake of clarity, the speculative mention that this TA system originated from *P. sordellii* has been removed from the discussion.

Lines 435-37, this statement seems to be missing a reference, is this also from ref 8?

Response :

This statement about TA in phiCD38-2 prophage is related to the general *in silico* search for TA systems within *C. difficile* prophages that we have performed in the present work. The sentence has been modified to clarify this point (lines 489 to 490).

Figure 2 legend (lines 643-60), indicate more clearly in either the figure or the legend that the antitoxin is detected with the antitoxin probe in 2A but the riboswitch probe in 2B. For the stability experiments of Fig 2C, state in either the legend here or the materials and methods how many replicates the half-lives were calculated from.

Response :

As suggested by the Reviewer, we clarified in both the figure and the figure legend (line 701) which probe was used to detect the antitoxin in the different Northern blots. For the sake of clarity, we have also included a schematic representation of probe position for detection of different transcripts to Figure 2A.

The number of replicates has been included into the figure legend: “The half-lives for toxin and antitoxin transcripts have been estimated from three independent experiments. » (lines 704-705).

Figure S3 legend, line 735 – “Potential processing site is shown by vertical green arrow” but there isn’t a vertical green arrow in the figure, and line 739, “in blue boxes” instead of “underlined in blue” for consistency

Response :

There was a vertical green arrow but we agree it was hard to see, we have increased the size of the green arrow in the figure to facilitate the figure analysis. “in blue boxes” has been added (line 800).

Figure 3, indicate variation in the growth experiments by adding standard deviation error bars to the growth curves

Response :

Done

Figure S4 legend (as for Fig 2C) state how many replicates the transcript half-lives were calculated from

Response :

The number of replicates has been included to the Figure S4 legend (Now Fig. S5 in revised version) (lines 822-823).

Figure S7 legend, specify whether this is the short or long transcript (and ideally include structure prediction for short and long versions, as in major comment 1)

Response :

The structure predictions have now been performed for both the short and long forms of antitoxin transcripts. The figure and the legend (line 836) have been modified accordingly (now Figure S8 in revised version).

Figure S10D and 11, use correct Δ symbol in horizontal axis labels

Response :

This has been corrected

Figure S13, what do the red dots by two of the toxin homologs signify?

Response :

Red dot with tag “small variant 1” indicates CD0904.1

Red dot with tag “small variant 2” indicates CD0956.3

This information has been included into the Figure legend (now Figure S14 in revised version) (lines 885-886).

Reply to comments of Reviewer 3

We would like to thank the Reviewer for the constructive criticism and helpful suggestions for improving the manuscript.

Reviewer #3 (Remarks to the Author):

There are few comments that the authors can attempt to address in their revised manuscript.

1. For growth Curves, CFU analysis and light microscopy why only 1 TA pair (CD0977.1-RCd11) was chosen (Refer to S2ABC)

Response:

All type I TA modules identified in this study shared extensive sequence homology. As explained in the manuscript, we have chosen *CD0977.1-RCd11* TA module present in both prophages of CD630 strain for more detailed analysis because of its association with c-di-GMP-responsive riboswitch that was unique and intriguing.

2. According to me CFU analysis data should be moved to main figures.

Response:

We have placed these data on CFU analysis into the supplementary material due to the space limitations for paper format. The number of main figures accepted being limited we suggest to keep this analysis as supplementary figure.

3. Author think 10 % longer cells can be significant number, Discussion lacks explanation behind how toxin expression can result in increased cell length. Also, why this phenomenon is only seen in 10% cells? Can the authors explain this?

Response:

Toxins from TA modules in other Gram-positive bacteria have been reported to affect cell envelope biosynthesis, cell division and chromosome segregation, however, the mechanisms remain to be uncovered. In our previous work with other type I TA modules in *C. difficile* we have already noticed similar changes in cell length for 5-9% of cells. Such cell heterogeneity in TA module expression and effects could be expected from other TA module investigations. Further studies would be needed to provide explanation of these observations.

4. The authors can shift bioinformatics analysis part before the results for characterization of Type I TA systems.

Response:

We have placed the bioinformatics prediction part at the end of the results section to provide the data on the general prevalence of these TA systems in *C. difficile* phages.

5. Can the authors functionally characterize few toxins (3-4) identified from bioinformatic work (fig. 6) in growth inhibition and cell length analysis.

Response:

The characterization of additional toxins from phages lies beyond the scope of this study. Together with our previous work (Maikova et al 2018) we have provided the experimental data for 7 type I TA modules in *C. difficile*.

6. Figure 3 legends: Line 666: How many experiments sets were done is missing

Response:

This has been added: Values represent means \pm standard deviations ($N = 3$) (lines 717-718).

7. Axis label in Fig. 5C should be replaced with % phiCD630-1 maintenance.

Response:

The axis label has been modified as suggested.

8. The methods sections can be shortened.

Response: To shorten the methods section, details of vector construction have been removed from the main text and transferred in Supplementary Methods.

9. The authors have referred to various strains which were published earlier. A brief explanation of these strains-a line each justifying their use would be beneficial for general

audience.

Response: Brief explanations have been added where appropriate.

line 178-179: by expressing the gene *dccA*, coding for a diguanylate cyclase involved in c-di-GMP production, from a plasmid

line 218-219 : depletion of the RNA chaperone protein Hfq, which generally increases the intracellular half-life of sRNAs and stabilizes the interactions between sRNAs and their target mRNAs

line 223-224 : for the ribonucleases RNase III, RNase J and RNase Y, that could be involved in toxin and antitoxin RNA decay.

Reply to comments of Reviewer 4

We would like to thank the Reviewer for the constructive criticism and helpful suggestions for improving the manuscript.

Reviewer #4 (*Remarks to the Author*): *The manuscript by Peltier et al. describes the identification of new type I TA systems belonging to prophage regions of C. difficile. The authors show that toxins from these modules can be used as efficient counter-selectable marker for chromosome manipulation in C. difficile, and they provide the first evidence that type I TA systems could be involved in prophage maintenance.*

Minor points:

-Toxin names such as CD0977.1 make the result and figure parts difficult to follow. It would be easier if the authors could simplify their nomenclature

Response:

We have kept the standard nomenclature for the toxin gene names in *C. difficile* used by the community working on this bacterium. For new genes of antitoxins we have introduced new RNA names RCdX.

-Fig 2A and 2B are not clear. How the RNA loads were adjusted (specially when the toxin was overexpressed)? What are the longer transcripts detected in pT and pTA lanes of Fig2A for the toxins? The relative abundance of transcripts seems to be gel-related. Are the transcripts detected with an antitoxin probe around 150 Nt Fig2A and 140 Nt Fig2B the same, and why results are so different with antitoxin probes between Fig2A and Fig2B? The authors could add a schematic to summarize the different transcripts starts and ends.

Response:

The RNA loads have been adjusted to the total RNA amounts, 5S RNA serves as loading control and is presented at the bottom of each Northern blot. The longer transcripts derive from readthrough of transcription on the plasmid after ATc promoter induction. Figure S1 shows a schematic of TA locus transcription and Figure S3 presents the sequence of the TA locus with all annotated transcript forms, transcript 5' and 3' ends and regulatory elements. The origin of different transcripts was described in the initial version of the manuscript. For antitoxin, 150-nt transcript is originated from P2 downstream promoter, while 140-nt transcript corresponds to a prematurely terminated riboswitch transcript from P1 promoter. The Northern blot for antitoxin detection in Fig. 2B (now Fig. 2C) was mislabelled making the analysis of the result unclear and this has now been modified. With antitoxin

probe used for Fig. 2A (now Fig. 2B), we can detect short 150-nt transcript, long 400-nt transcript and 300-nt processed form. With riboswitch probe used for Fig. 2B (now Fig. 2C) we could detect the long full-length 400-nt transcript and a terminated 140-nt transcript. To clarify the position of probes with respect to identified transcripts, we have now included the schematic representation of different transcripts and probes for their detection in Figure 2A.

-page 7, lanes 166-167: The authors should clarify the reverse correlation they observe.

Response :

As requested also by Reviewers 1 and 2, this statement has been clarified and additional details on the relative toxin and antitoxin levels have been included in the revised version of the paper as follows:

Lines 170-176: « The toxin overexpression in the presence of ATc inducer resulted in a decreased amount of the major 150-nt RCd11 antitoxin expressed from chromosomal location (lanes “pT” compared in the absence and in the presence of ATc). Similarly, for the strain carrying the entire TA locus on pTA plasmid expressing the antitoxin from its own strong promoter, the toxin overexpression after ATc induction led to a decrease in the 150-nt RCd11 antitoxin level (lanes “pTA” compared under conditions “-ATc” and “+ATc”).»

-For results presented in Fig3A, the authors could provide northern blots to show that mutated antitoxin transcripts are still expressed.

Response :

We generated a series of promoter fragments fused to the *phoZ* alkaline phosphatase reporter gene to verify transcription from our different constructs. Alkaline phosphate activity was observed for all fusions, suggesting that antitoxin transcripts are still expressed. Results are presented in Fig. 3B and S6B.

-Overall there are a lot of technical details in the results part that could be taken out for clarification.

Response :

We decided to keep the details in the results section that could facilitate the reading of the paper.

-I do not understand why pMSR0 vector can be used for allele exchange in C. difficile 027

Response :

As indicated in Materials and Methods section, the pMSR0 vector carries the RCd8 antitoxin to cognate toxin CD2517.1 due to the absence of this gene in the chromosome of *C. difficile* 027 ribotype strain.

“It should be noted that expression of the RCd8 antitoxin from the pMSR0 vector was required to counteract the basal expression of CD2517.1 toxin gene due to the *Ptet* leakiness. In contrast, expression of the RCd8 antitoxin from the pMSR vector was not required since the *CD2517.1-RCd8* TA module is naturally present within the chromosome of *C. difficile* 630. Native expression of RCd8 was therefore sufficient to prevent CD2517.1 production from the plasmid”.

Changes made to Figures:

All changes in figures are specified here and all updated figures are shown below

Fig. 2:

- A schematic representation of different transcripts and probes for their detection has been included in Figure 2A.
- The Northern blot for antitoxin detection in Fig. 2B (now Fig. 2C) was mislabelled and this has now been modified (now reads “riboswitch RCd11/RCd12”).
- +/- *pdccA* Northern blot with the probe that detects the short transcript of the antitoxin (AT RCd11/RCd12) has been added in Fig 2C.

Fig. 3:

- Standard deviation error bars have been added to the growth curves
- Results of alkaline phosphatase fusions to measure the P1 and P2 promoter activities under various conditions are now presented in Fig. 3B.

Fig. 5:

- We have now added the time point 16 h in Fig. 5B to show that the strains are still growing between 16 and 24 h.
- Axis label in Fig. 5C has been modified.

Fig. S3:

- The size of the green arrow has been increased.

Fig. S4:

- This is a new figure presenting the analysis of TA RNA duplex formation by RNA band shift assay.

Fig. S6 :

- Results of alkaline phosphatase fusions to measure activity of the RCd11 promoter P_1 + disrupted P_2 are now presented in 6B.

Fig. S8 :

- The predictions for long and short forms of RCd11 antitoxins are now shown.

Fig. 2

Fig. 3

A

B

C

Fig. 5

A

B

C

Fig. S3

Fig. S4

Fig. S6

A

B

C

Fig. S8

Reply to comments of Reviewer 2

We would like to thank the Reviewer for the constructive criticism and helpful suggestions for improving the manuscript.

All the minor points raised by Reviewer were considered and the suggested modifications were made.

Reviewer #2: Remarks to the Author:

Summary

The authors have revised their previously submitted manuscript on novel type I TA systems of Clostridioides difficile and their contribution to prophage maintenance with extra experiments, clarifying elements added to the figures and re-writing of sections of the text. Overall they have done an excellent job. The extra experiments, in particular the in vitro RNA interaction EMSAs and the promoter activity assays, significantly strengthen the manuscript. The schematics of northern blot probe positions and expected transcript lengths that they detect (Fig 2A) and promoter regions used for the alkaline phosphatase reporter assays (Fig 3B) are very helpful for interpretation of the results. The findings are important for the field and will certainly advance the state of the art regarding type I TA system function and regulation in general, not only in this species. I commend the authors for their efforts and have only a few minor comments, below.

Minor comments

1. line 183, c-di-GMP (not ci)

Response :

Done

2. Lines 185-87, the stated 20x differential expression of this toxin in biofilm-grown cells relative to planktonic by qRT-PCR is still in good support of the authors' conclusions about the biological relevance of the TA system even if no c-di-GMP effect was seen, and as such it merits showing the data. Not necessarily as a new figure, but in either a small table or directly in the main text including information on 1) how many replicates were performed, 2) average and standard deviation of the values if $n > 1$ and 3) whether it was normalized to a control transcript and if so, what was it. No objection to leaving the negative result for the antitoxin as "data not shown" statement.

Response :

As suggested we have now added the supporting results in the text "20.4 ± 5.0, N=3 biologically independent samples, with 16S rRNA gene for normalisation" and removed "data not shown" statement.

3. Line 269-71, it is stated that the promoter-reporter containing P1 + disrupted P2 still retained AP activity, suggesting that the nucleotide substitutions in P2 (of the TSS and the -10 box, as per the supplementary methods) do not prevent expression of the antitoxin. But in Fig S6B this activity is only about 12 AP units. This is extremely low compared to the values in the graph of Fig 2B which look like about 80 units for P1 alone, 350 for P2 and maybe 500 units for P1 + P2 (in exponential growth). Moreover, comparing Fig 3A with Fig S6B the pTA construct pDIA6817 containing the Ptet-toxin and the P1(+ disrupted P2) antitoxin is incapable of allowing growth on BHI-ATc plates with 10 or 100 ATc, but it can partially restore growth on 5 ATc. So this provides an explanation for why the growth of the strain containing pDIA6817 is partially restored on 5 ng/ml ATc (cf lines 259-61) – this low level of

antitoxin expression from the P1 + disrupted P2 can neutralize the toxin when it is weakly expressed, but overwhelmed when more ATc inducer is included in the plates. Suggest rewriting for better synthesis of the results.

Response :

We have followed the reviewer recommendation and now modified the text as follows:
“Of note, the promoter fragment comprising P_1 and the disrupted P_2 retained **low** AP activity (Supplementary Fig. 8b), suggesting that the nucleotide substitutions in P_2 do not **completely** prevent expression of the antitoxin transcripts. **Low level of antitoxin expression from P_1 promoter could thus explain the partial restoration of growth only in the presence of lower dose of ATc (5 ng/ml) when toxin is weakly expressed (Supplementary Fig. 8a).**”

4. Minor inconsistency between legends of figure 3 and figure S6. Figure 3 legend states that 4 hr growth is exponential phase (line 719) and figure S6 legend states that 4 hr growth is stationary (line 830).

Response :

We have modified the Supplementary figure S6 legend (now in Supplementary information single file) and replaced “stationary” by “**exponential**”.